# An interpretable approach to estimate the self-motion in fish-like robots using mode decomposition analysis

Yufan Zhai ⬭[1], Xingwen Zheng[2,3], Li-Ming Chao[4,5,6], Shikun Li[1], Minglei Xiong[1], Yongxia Jia[7], Liang Li ⬭[4,5,6,8] ✉ & Guangming Xie ⬭[1,9] ✉

The artificial lateral line system, composed of velocity and pressure sensors, is the sensing system for fish-like robots by mimicking the lateral line system of aquatic organisms. However, accurately estimating the self-motion of the fish-like robot remains challenging due to the complex flow field generated by its movement. In this study, we employ the mode decomposition method to estimate the motion states based on artificial lateral lines for the fish-like robot. We find that primary decomposed modes are strongly correlated with the velocity components and can be interpreted through Lighthill's theoretical pressure model. Moreover, our decomposition analysis indicates the redundancy of the sensor array design, which is verified by further synthetic analysis and explained by flow visualization. Finally, we demonstrate the generalizability of our method by accurately estimating the self-states of the fish-like robot under varying oscillation parameters, analyzing three-dimensional pressure data from the computational fluid dynamics simulations of boxfish (*Ostracion cubicus*) and eel-like (*Anguilla anguilla*) models, and robustly estimating the self-velocity in complex flows with vortices caused by a neighboring robot. Our interpretable and generalizable data-driven pipeline could be beneficial in generating hydrodynamic sensing hypotheses in biofluids and enhancing artificial-lateral-line-based perception in autonomous underwater robotics.

Animals have evolved advanced sensory and motor capabilities that allow them to effectively navigate natural environments, and these biological strategies have inspired the development of underwater robots[1]. As a revolutionary method for engineering, biomimetic technology involves understanding biological principles in animals to enhance the abilities of robots in perception, mobility, and decision-making[2,3]. Robotic engineers incorporate the body morphologies and physiological structures of animals into their designs, attempting to endow robots with the capability to robustly work in complex environments[4]. These robots can possess several desirable properties in animals, including adaptability, flexibility, and agility[5]. Fish-like robots, which mimic the morphology, kinematics, and dynamics of real fish, are one of the typical examples[6,7].

[1]State Key Laboratory for Turbulence and Complex Systems, Intelligent Biomimetic Design Lab, College of Engineering, Peking University, Beijing 100871, China. [2]Institute of Cyber-Systems and Control, Department of Control Science and Engineering, Zhejiang University, Hangzhou 310027, China. [3]State Key Laboratory of Ocean Sensing, Zhejiang University, Zhoushan 316021, China. [4]Department of Collective Behaviour, Max Planck Institute of Animal Behavior, Konstanz 78464, Germany. [5]Centre for the Advanced Study of Collective Behaviour, University of Konstanz, Konstanz 78464, Germany. [6]Department of Biology, University of Konstanz, Konstanz 78464, Germany. [7]School of Aerospace Engineering, Tsinghua University, Beijing 100084, China. [8]Department of Computer and Information Science, University of Konstanz, Konstanz 78464, Germany. [9]Institute of Ocean Research, Peking University, Beijing 100871, China. ✉e-mail: lli@ab.mpg.de; xiegming@pku.edu.cn

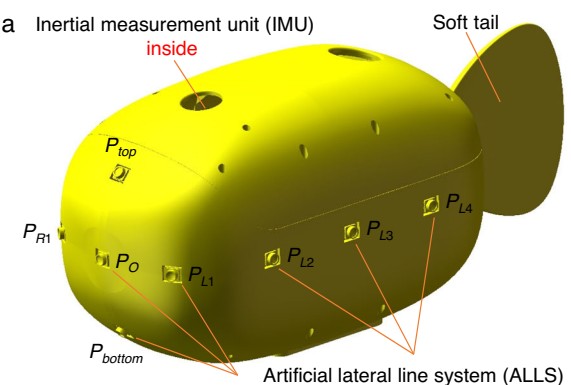

**Fig. 1 | The experimental platform for the free-swimming fish-like robot (detailed in Supplementary Movie 1). a** The fish-like robot is equipped with 11 uniformly distributed ALLS sensors and an inertial measurement unit as well. **b** The fish-like robot swims freely in a tank with still water and the trajectory is recorded by a top-view camera.

In real underwater applications, where light is often insufficient for vision and acoustic sensors frequently suffer from interference from natural sources like currents, marine life, and other ambient noise, fish-like robots require other types of sensors to acquire information on both environments and self-states efficiently and robustly. The lateral lines of aquatic organisms could be an ideal solution, as fish have successfully used this organ to perceive sufficient information from the flow velocity and pressure field nearby[8] to navigate, forage, and thrive[9,10]. Inspired by this, various kinds of artificial lateral line systems (ALLS) consisting of velocity or pressure sensors have been developed and served as the sensing systems of fish-like robots[11-15]. For instance, ALLSs have been used for motion control[16-20], perception of vortices[21,22] and dipole sources[23-27], obstacle avoidance[28-30], and perception among schools of fish-like robots[31-35].

Estimating the self-states, such as the velocity and trajectory, is crucial for both biological and robotic systems[36,37]. Compared with conventional underwater robots, the oscillatory motion of a free-swimming fish-like robot introduces a complex fluid-structure interaction issue, which makes the estimation more difficult. To solve this, the existing estimation methods can be broadly divided into two categories. The first category is based on qualitative relationships in fluid dynamics, such as Bernoulli's equation[16,36], and uses simple regression models for estimation. Nevertheless, these models only focus on the correlations between the average pressure and motion states, and cannot interpret the underlying causes of pressure variations. The second category employs neural networks to analyze ALLS data[38]. A significant drawback of neural networks is their lack of interpretability, which hampers their applicability to a broader range of problems, such as adapting to fish-like robots with different morphologies.

In addition, another challenge lies in arranging ALLS sensors on fish-like robots. Most current solutions adopt either a straight-line or uniform configuration around the body, which is oversimplified compared with natural lateral line systems. Optimizing the locations of ALLS sensors remains in its infancy[39-42], with no generalized method or paradigm established for addressing this issue.

Recently, mode decomposition methods have been used to analyze the variations in flow fields, such as Kármán vortex streets behind a cylinder in the coming flow[43-46]. This kind of method can extract the main components in space, which are known as modes, and determine the coefficients of temporal evolution. Furthermore, it can provide a reduced-order model (ROM) for the flow field and help to gain a deeper understanding of the flow mechanisms. Based on mode decomposition methods, ALLS can be enhanced[47] and used for flow field reconstruction[48] and underwater robot control[49].

In this paper, we explore whether, if so, how mode decomposition methods could be applied in estimating the self-states of fish-like robots using ALLS (detailed in Supplementary Note 1 and Supplementary Fig. 1). We collect the spatiotemporal pressure data using ALLS when the fish-like robot swims freely in the rectilinear and turning motions. With mode decomposition analysis, we discover that pressure variations on the surface can be decomposed into several modes, which are strongly correlated with the motion of the fish-like robot, such as the forward motion and the oscillating motion. The components of pressure variations from mode decomposition can be interpreted through Lighthill's theoretical pressure model. The dimension-reduced representation of ALLS data also shows the redundancy of our ALLS design and predicts the specific number and locations of sensors for sufficient self-state estimation. Further synthetic analyses verify the prediction, and flow visualization explains it well. We finally demonstrate the generalizability of our method in three complex cases. (1) Our method works effectively for the free-swimming fish-like robot under various oscillation parameters, including varying frequencies, amplitudes, and offsets. (2) It can be extended to three-dimensional pressure data obtained through three-dimensional computational fluid dynamics (CFD) simulations with both boxfish and eel-like models, suggesting that our method can be generalized to fish-like robots with different morphologies and swimming styles. (3) The estimation method proves robust in self-velocity estimation of the fish-like robot swimming in complex flows with vortices shedding from a neighboring robot. Overall, our interpretable and generalizable data-driven method showcases its strong capability in estimating the self-states of autonomous fish-like robots across various conditions and configurations.

## Results

### Interpretable mode decomposition

The fish-like robot (Fig. 1a and detailed in the "Methods" section) is inspired by boxfish (*Ostracion cubicum*) in nature. Its density is slightly less than one so that almost the entire body is submerged below the water surface, except the antenna used for communication with the computer. The fish-like robot has a system inside for changing the center of gravity, which is used for adjusting the pitch and roll angles to around zero before swimming. Therefore, it can swim forward and turn freely within a two-dimensional plane at a consistent depth by adjusting the oscillation amplitude, frequency, and offset of the soft tail. The fish-like robot is also equipped with an inertial measurement unit (IMU) and ALLS consisting of 11 pressure sensors for pressure collection. All the sensors except $P_{top}$ and $P_{bottom}$ are located in a horizontal plane. Thus, we can obtain adequate pressure data in the

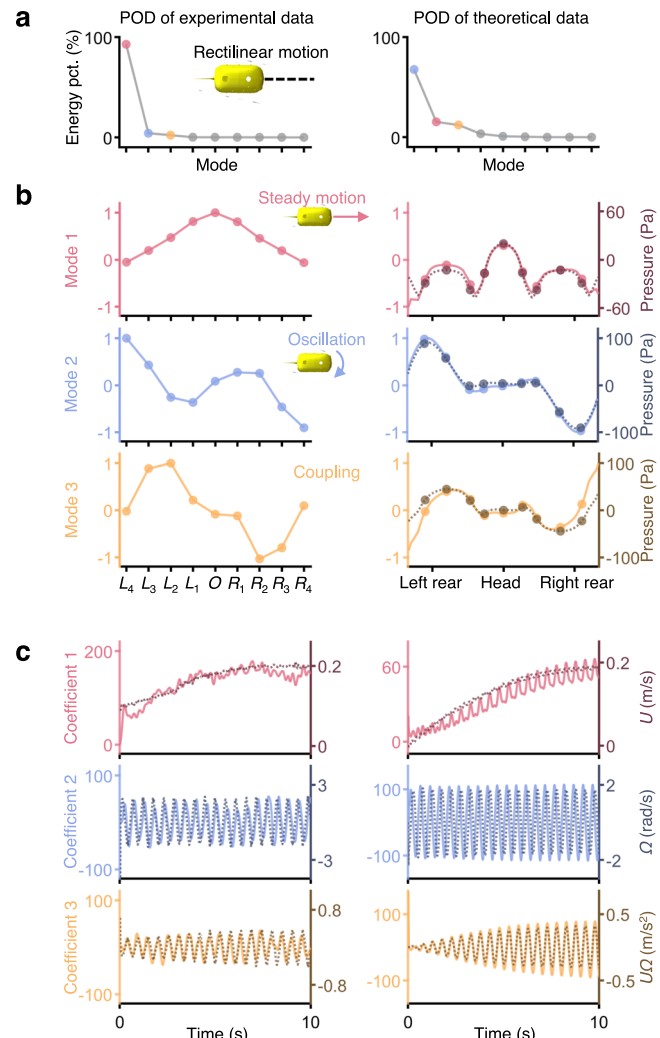

**Fig. 2 | Mode decomposition (POD) results of the hydrodynamic pressure data in the rectilinear motion.** The parameters of tail oscillation in the example are frequency = 2 Hz, amplitude = 30°, and offset = 0°. The first column represents the POD results of experimental data. The second column represents the POD results of theoretical data from the panel method. **a** The energy distribution across each mode in POD shows that the first three modes account for nearly all the energy. **b** The first three dominant modes (solid lines), as a function of pressure sensor locations, can be interpreted as representing the pressure caused by steady motion (red dashed lines), oscillation (blue dashed lines), and various coupling motions (orange dashed lines), respectively. The mode values are normalized such that the maximum absolute value is one. **c** The coefficients of modes (solid lines), as a function of time, are interpreted as variations in forward velocity (red dashed lines), angular velocity (blue dashed lines), and coupling terms (orange dashed lines).

two-dimensional plane and analyze the impact of sensor locations on the perception. Since only two sensors ($P_{\text{top}}$ and $P_{\text{bottom}}$) are out of the plane, we lack pressure data to analyze the distribution of sensors in the third dimension. Therefore, in the follow-up experiments, we only consider the rectilinear and turning motions of the fish-like robot and evaluate the nine pressure sensors in the horizontal plane for self-state estimation.

The experiments of an individual free-swimming fish-like robot are conducted in a tank ($3 \times 2 \times 0.8$ m) containing still water (0.5 m in height) (Fig. 1b and detailed in the "Methods" section). The trajectory of the fish-like robot is recorded by a top-view camera. The pressure data collected by ALLS are analyzed by a typical mode decomposition method named Proper Orthogonal Decomposition (POD), which

projects a high-order and nonlinear system into a low-dimensional space through an orthogonal basis (detailed in Supplementary Note 2)[43].

The pressure data is expressed as

$$\mathbf{P} = \begin{bmatrix} \mathbf{p}(t_1) & \mathbf{p}(t_2) & \cdots & \mathbf{p}(t_M) \end{bmatrix} \quad (1)$$

where $\mathbf{p}(t_i) \in \mathbb{R}^N$ represents the snapshot of $N$ sensor points at time $t_i$. $M$ represents the length of sampling time. The spatiotemporal data can be decoupled by POD, which is shown as

$$\mathbf{p}(t_i) = \sum_{j=1}^{r} a_j(t_i)\mathbf{u}_j(\mathbf{x}) \quad (2)$$

where $\mathbf{x}$ represents the spatial locations of pressure sensors in the body coordinate frame and $\mathbf{u}_j(\mathbf{x}) \in \mathbb{R}^N$ represents the basis of POD, whose norm equals one, namely the mode. $a_j(t_i)$ represents the coefficient of the $j$th mode at time $t_i$. $r$ represents the order of ROM, which means that we can analyze the system in a low-dimensional space. The experimental pressure (at the sampling rate of 50 Hz) in the rectilinear motion is decomposed into several time-invariant modes with time-varying coefficients, and the modes have different energy proportions (the first column in Fig. 2). 

To interpret the dominant modes in POD, we establish a two-dimensional theoretical model that describes the hydrodynamic pressure variations on the surface of a swimming fish-like robot based on Lighthill's pressure model[50] (detailed in Supplementary Note 3). Under the assumption of irrotational flow, the pressure variations on the surface can be expressed as

$$\begin{aligned} \frac{p}{\rho} &= -\frac{\partial \varphi}{\partial t} - \frac{1}{2}\|\nabla\varphi\|^2 \\ &= C_1 U^2 + C_2 V^2 + C_3 \Omega^2 + C_4 UV + C_5 U\Omega + C_6 V\Omega \\ &\quad + C_7 \frac{dU}{dt} + C_8 \frac{dV}{dt} + C_9 \frac{d\Omega}{dt} \end{aligned} \quad (3)$$

where $U, V, \Omega$ represent the velocity and angular velocity in the body coordinate frame. $\rho$ represents the density of water. $C_i$ only depends on the body morphology and is independent of time or velocity. We apply panel methods[51] with a given velocity component $U$, $V$, or $\Omega$, respectively, to calculate the coefficients $C_i$ for the main components in Lighthill's pressure model (detailed in Supplementary Note 3). We then generate pressure data with $U$, $V$, and $\Omega$, but this time coupling like swimming freely, and apply POD to the pressure data. We find that the three main modes extracted from POD correspond to the coefficients in Lighthill's pressure model (the second column in Fig. 2). Specifically, for the rectilinear motion, mode 1 aligns well with $C_1$ of $U$ in Lighthill's pressure model, which represents the pressure resulting from the steady motion forward. And the corresponding coefficient 1 in POD varies consistently with the forward velocity. Similarly, we compare modes 2 and 3 extracted from POD with the pressure caused by angular velocity $\Omega$ and coupling term $U\Omega$. The modes show strong matches with the theoretical pressure distributions caused by the decomposed swimming patterns. And the coefficients reflect the varying velocity components. These findings suggest that POD can effectively decompose the pressure data into several main modes and coefficients, which are interpretable within the framework of Lighthill's pressure model.

For the experimental data, we also find the main modes extracted by POD qualitatively match the coefficients $C_i$ from Lighthill's pressure model as well (the first column in Fig. 2), although there are some quantitative differences because the assumption of potential flow theory is different from the actual environments and the two-

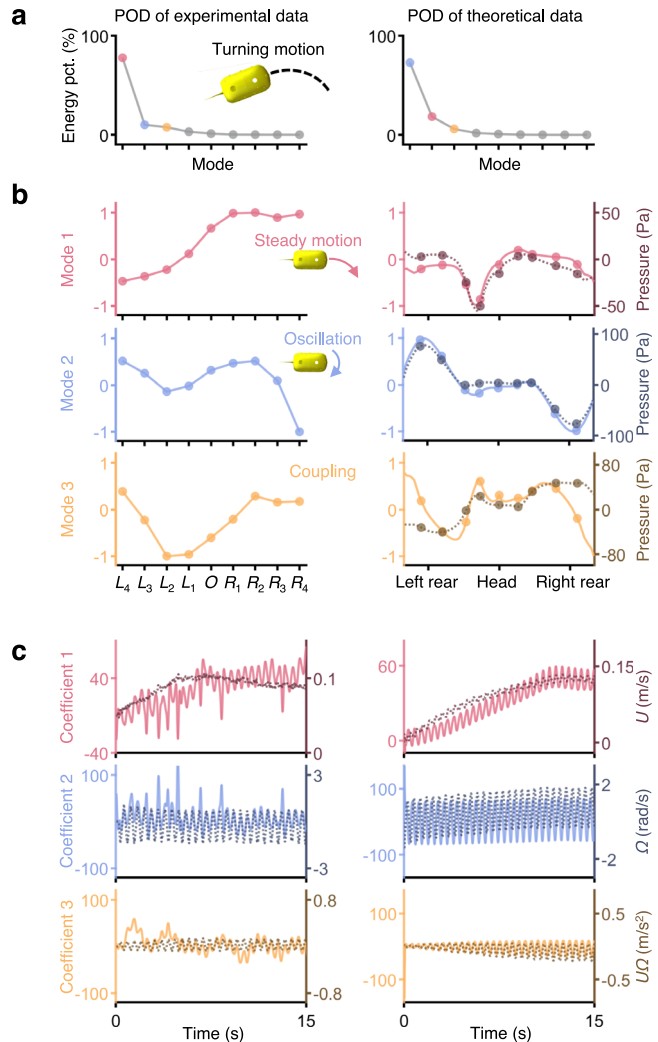

**Fig. 3 | Mode decomposition (POD) results of the hydrodynamic pressure data in the turning motion.** The parameters of tail oscillation in the example are frequency = 2 Hz, amplitude = 20°, and offset = 20°. The first column represents the POD results of experimental data. The second column represents the POD results of theoretical data from the panel method. **a** The energy distribution across each mode in POD shows that the first three modes account for nearly all the energy. **b** The first three dominant modes (solid lines), as a function of pressure sensor locations, can be interpreted as representing the pressure caused by steady motion (red dashed lines), oscillation (blue dashed lines), and various coupling motions (orange dashed lines), respectively. The mode values are normalized such that the maximum absolute value is one. **c** The coefficients of modes (solid lines), as a function of time, are interpreted as variations in forward velocity (red dashed lines), angular velocity (blue dashed lines), and coupling terms (orange dashed lines).

dimensional Lighthill's pressure model is different from the three-dimensional fish body (detailed in Supplementary Note 3). Specifically, mode 1 (red lines in Fig. 2b) matches the pressure distribution when the fish-like robot moves forward steadily without oscillation, which equals $C_1 U^2$ according to Eq. (3). The pressure distribution of mode 1 exhibits apparent left-right symmetry, with the highest value at the head serving as the stagnation point. The variation of the corresponding coefficient (red lines in Fig. 2c) is similar to the swimming velocity $U$, showing that the fish-like robot gradually accelerates to a constant velocity. Mode 2 (blue lines in Fig. 2b) reflects the pressure generated by the in-place oscillation, which is caused by $\Omega$ in Eq. (3). Two local maxima are located at the left rear and right front of the body, resulting from the oscillation of the fish-like robot, with the head rotating and pushing water away. Due to the periodic oscillation of the

fish-like robot, the corresponding coefficient (blue lines in Fig. 2c) exhibits periodic variations. Mode 3 (orange lines in Fig. 2) reflects the coupling of steady motion and oscillation (detailed in Supplementary Note 4). The remaining modes omitted in Fig. 2b, c may reflect other terms in Eq. (3). However, they are less important from the perspective of energy (Fig. 2a).

Thus, the decomposition of pressure variations by POD can be interpreted through Lighthill's theoretical pressure model. Modes $\mathbf{u}_j(\mathbf{x})$ in Eq. (2) reflect $C_i$ in Eq. (3). The decomposed modes from different examples are almost the same (detailed in Supplementary Note 6), which is consistent with the only dependence of $C_i$ on the body morphology. Coefficients $a_j(t_i)$ in Eq. (2) reflect velocity components in Eq. (3), which both vary over time. The energy of the modes depends on which parameter is dominant while the fish-like robot is swimming. For example, in the rectilinear motion, the energy of the mode corresponding to $U$ is the highest.

Furthermore, coefficient 1 and the swimming velocity vary consistently, suggesting that the fish-like robot is dynamically converging to the stable swimming velocity (red lines in Fig. 2c). Since there is a quadratic relationship between the pressure variations and the velocity in Lighthill's pressure model (Eq. (3)), we train a quadratic regression model from experimental data and propose a method using this model to estimate the velocity of the fish-like robot (detailed in the "Methods" section and Supplementary Note 5). Besides, concerning the energy distribution, the first several (about three) modes account for almost all of the energy (Fig. 2a). The pressure data collected by ALLS can be represented in a lower dimension, indicating the redundancy of our ALLS design with nine sensors. In the following sections, we will show that the estimation of self-states can be enhanced from these two aspects which are derived from POD.

We further conduct experiments involving fish-like robots swimming in the turning motion and perform the same mode decomposition analyses (Fig. 3). The pressure variations are also decomposed into several modes with coefficients which are strongly correlated with the self-motions. The first three modes are still dominant and account for nearly all of the energy (Fig. 3a). Compared with the rectilinear motion, the main difference is that mode 1 is asymmetrical since the fish-like robot is turning right (Fig. 3b). Therefore, mode 1 is related to the steady motion which includes moving forward and turning. The location with higher pressure is on the right head part which serves as a front stagnation point and pushes the water away. We conduct 8 additional cases under different oscillation parameters, and the results show that our method can effectively decompose the pressure data into three main interpretable components like above (detailed in Supplementary Note 6).

## Optimizing the number and locations of sensors for self-trajectory estimation

For autonomous fish-like robots, estimating their own trajectory is more practically significant than estimating their self-velocity[36,37]. Given that POD demonstrates potential for estimating the self-velocity, we can accurately estimate the trajectory of the fish-like robot by integrating the velocity estimated by POD and angular velocity measurements from the IMU (detailed in the "Methods" section). For comparison purposes, we also perform trajectory estimation using the regression method described in our previous work[36].

Trajectories of a free-swimming fish-like robot can be estimated via nine sensors in a horizontal plane. However, the low-dimensional representation of ALLS data based on POD indicates that it may not be necessary to use all nine sensors. Due to the different flow structures at different locations on the surface, some of the sensors may have a negative impact on perception. According to the energy distribution of modes (Figs. 2a and 3a), the ideal number of sensors could be about three. Furthermore, the sensors can be selected according to the value of mode 1 from POD which corresponds to the swimming velocity $U$.

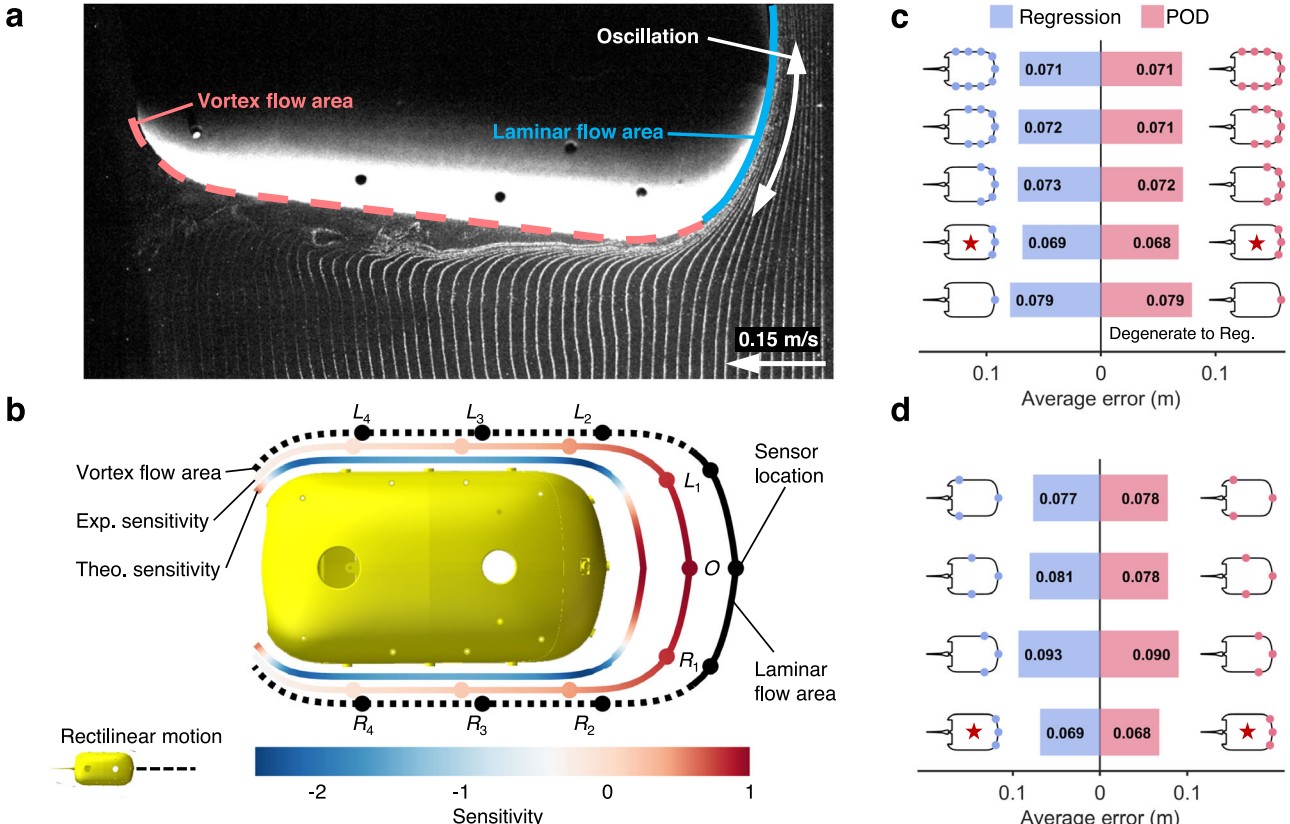

**Fig. 4 | Mode decomposition (POD) optimizes the number and locations of sensors for self-trajectory estimation in the rectilinear motion.** The parameters of tail oscillation in the example are frequency = 1.7 Hz, amplitude = 25°, and offset = 0°. **a** A screenshot of flow visualization by the hydrogen bubble technique in Supplementary Movie 2, which shows the laminar flow area and the vortex flow area on the surface of the fish-like robot. Considering the left-right symmetry, only half of the part is presented. **b** Evaluation of the locations of sensors on the surface. From outside to inside, the sequence is flow structure characteristics, 'experimental sensitivity', 'theoretical sensitivity', and the morphology of the fish-like robot. Solid points represent the sensors. **c**, **d** The average estimation errors of the free-swimming trajectory using different combinations of sensors. Red bars and blue bars represent the results by the POD and regression methods, respectively. If only selecting one sensor, the POD method degenerates into the regression method and the results are the same. Curves and solid points near the bars represent the fish-like robot and the sensors used in the estimation. The star represents the optimal combination of pressure sensors.

Since this mode accounts for the majority of the energy, the pressure in Eq. (3) is mainly determined by steady motion. For example, in the rectilinear motion, the pressure caused by steady motion is $C_1 U^2$. The value of the mode, namely $C_1$, reflects the variations of the hydrodynamic pressure when the swimming velocity varies. This is similar to the definition of sensor sensitivity, the ratio of output variations (pressure) to input variations (velocity). Therefore, we use the mode corresponding to $U$ to estimate the 'theoretical sensitivity' and 'experimental sensitivity' and check the accuracy with the sensitivity from traversal (detailed in Supplementary Note 7 and Supplementary Fig. 7).

In the rectilinear motion, it is indicated that the stagnation point at the head is the most sensitive to the velocity variations (Fig. 4b). More specifically, the 'experimental sensitivity' decreases as the distance to the head increases. Therefore, the three sensors $L_1$, $O$, $R_1$ can be potentially used for improving the trajectory estimation. The 'theoretical sensitivity' of the two sides decreases to a negative value lower than −1, which means the pressure variations are more significant but in the opposite direction. Since the oscillation of the fish body generates vortices and causes complex pressure variations, 'theoretical sensitivity', which comes from the theoretical model, may be inaccurate on both sides (detailed in Supplementary Note 3).

Actual trajectory estimation results using different combinations of sensors are used to validate the effectiveness of POD modes for sensor selection (detailed in the "Methods" section). The minimum error is reached by using three sensors $L_1$, $O$, $R_1$ for both methods

(Fig. 4c, d), which is consistent with the prediction based on POD. And surprisingly, the more sensors that are used, the higher the error. When compared with using all nine sensors, the trajectory estimation errors for the regression method and the POD method can be reduced by 3.6% (from 0.071 m to 0.069 m) and 4.1% (from 0.071 m to 0.068 m), respectively. The estimated trajectories are shown in Supplementary Note 7 and Supplementary Fig. 8.

To further understand this, we conduct flow visualization with the hydrogen bubble technique (detailed in the "Methods" section and Supplementary Note 9). From flow visualization, the surface can be divided into a laminar flow area and a vortex flow area (Fig. 4a and Supplementary Movie 2). The laminar flow area is situated at the head part. As the flow moves past the body's side parts, the boundary layer detaches from the shell, evolving into complex vortex structures. The optimal sensor combination ($L_1$, $O$, $R_1$) is located within the laminar flow area (Fig. 4b), as opposed to the vortex flow area, where the pressure sensors experience higher noise levels.

In the motion of turning right, 'experimental sensitivity' indicates that the right front part ($R_1$, $R_2$) is more sensitive to the velocity. 'Theoretical sensitivity' indicates that $O$ and $R_1$ could be the optimal sensors and the pressure variations in the right side part are negatively related to the velocity and less sensitive than the right front part (Fig. 5c). Trajectory estimation results based on two methods are shown in Fig. 5d. The optimal combination of sensors mostly appears on the right front part of the body, which validates the sensor selection from POD. The optimal number of the selected sensors is two ($O$ and

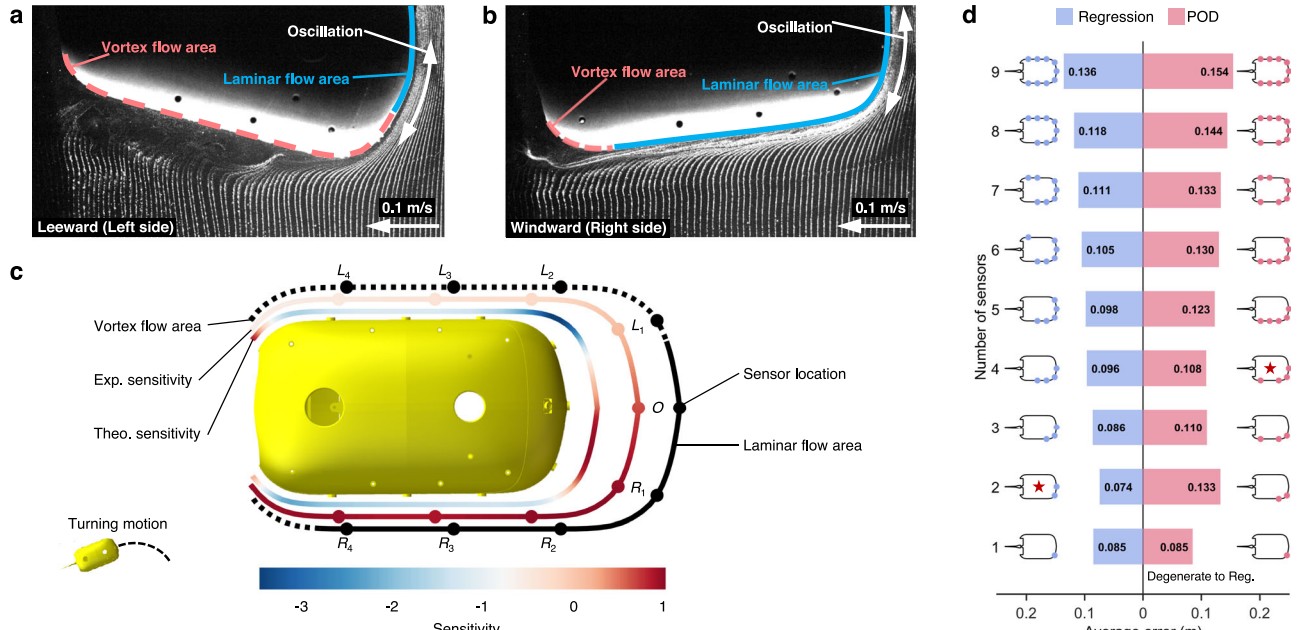

**Fig. 5 | Mode decomposition (POD) optimizes the number and locations of sensors for self-trajectory estimation in the turning motion.** The parameters of tail oscillation in the example are frequency = 1.9 Hz, amplitude = 20°, and offset = 15°. **a, b** Screenshots of flow visualization by the hydrogen bubble technique in Supplementary Movies 3, 4, which show the laminar flow area and the vortex flow area on the surface of the fish-like robot. **c** Evaluation of the locations of sensors on the surface. From outside to inside, the sequence is flow structure characteristics, 'experimental sensitivity', 'theoretical sensitivity', and the morphology of the fish-like robot. Solid points represent the sensors. **d** The average estimation errors of the free-swimming trajectory using different combinations of sensors. Red bars and blue bars represent the results by the POD and regression methods, respectively. If only selecting one sensor, the POD method degenerates into the regression method and the results are the same. Curves and solid points near the bars represent the fish-like robot and the sensors used in the estimation. The star represents the optimal combination of pressure sensors.

$R_1$) for the regression method and four ($O$, $R_1$, $R_2$, and $R_4$) for the POD method. The estimation errors of the regression and POD methods are decreased by 45.3% (0.136 m to 0.074 m) and 29.8% (0.154 m to 0.108 m) respectively. The estimated trajectories are shown in Supplementary Note 7 and Supplementary Fig. 9. If the number of the used sensors is about three, estimation errors are similar but much smaller than using all nine sensors. This is consistent with the energy distribution in Fig. 3a. The optimal combinations of sensors in 8 additional cases with different oscillation parameters also align with our reported results, shown in Supplementary Note 8.

We also conduct flow visualization for the turning motion and find the flow fields on both sides of the body are entirely different (Fig. 5a, b and Supplementary Movies 3 and 4). The leeward side (left side) almost belongs to the vortex flow area because the flow bypasses a large angle. The phenomenon of boundary layer separation is evident in this area. The windward side (right side) is almost in the laminar flow area, where the flow adheres to the surface, although the fish body is oscillating. The possible optimal sensors $O$ and $R_i (i = 1, 2, 3, 4)$ are all in the laminar flow area where the pressure variations are stable.

Sensor $R_4$, located at the rear part of the body near the vortex flow area, is included in the optimal combination (Fig. 5d). It may be because the 'experimental sensitivity' is high and the amplitude at the dominant frequency (the oscillation frequency) is the largest in the laminar flow area (detailed in Supplementary Note 7 and Supplementary Fig. 10). The pressure signal may be more significant due to the greater oscillation amplitude of the rear part, although vortices may exist. When facing noise, a larger amplitude of the effective signal means a larger signal-to-noise ratio, which will lose less information during signal processing.

### Generalizability of the method
The generalizability of our method has been demonstrated from three aspects. Firstly, we carry out the unsteady free-swimming experiments

of one fish-like robot under varying and dynamic oscillation frequencies, amplitudes, and offsets. For both rectilinear and turning motions, the fish-like robot is controlled to alternate between accelerating and decelerating. We find that the decomposed modes are similar to those cases with fixed oscillation parameters (Fig. 6a, b). Mode 1 and mode 2 correspond to the pressures caused by steady motion and oscillation, respectively, as described before. Coefficient 1 can capture the variations in forward velocity, including the acceleration and deceleration phases. And the amplitude and frequency of coefficient 2 vary as the oscillation parameters change. Therefore, the POD modes and coefficients are interpretable following Lighthill's pressure model, regardless of whether the oscillation parameters vary or not. While the fish-like robot is accelerating or decelerating, coefficient 1 exhibits larger fluctuations. It can also be interpreted by the acceleration term ($dU/dt$) in Lighthill's pressure model, which couples together with pressure caused by forward velocity in mode 1 (detailed in Supplementary Note 4). Despite such fluctuations, our proposed estimation method is still effective under varying oscillation parameters. For the rectilinear motion, the minimum error is reached by using three sensors $L_1$, $O$, $R_1$, rather than using all nine sensors. The estimated trajectory looks similar in Fig. 6c but using the optimal combination of sensors has smaller errors during the entire estimation process. The error is reduced by 14.8% (from 0.061 m to 0.052 m) for the regression method and 11.7% (from 0.060 m to 0.053 m) for the POD method. For the turning motion, the minimum error is reached by using three or four sensors on the right front part of the body. The error can be reduced by 33.6% (from 0.122 m to 0.081 m) for the regression method and 33.8% (from 0.154 m to 0.102 m) for the POD method (Fig. 6d, e). More unsteady cases with similar results are shown in Supplementary Note 8. Overall, our method for interpreting the decomposition and selecting the optimal combination of sensors to estimate the self-trajectory maintains its effectiveness under varying oscillation parameters.

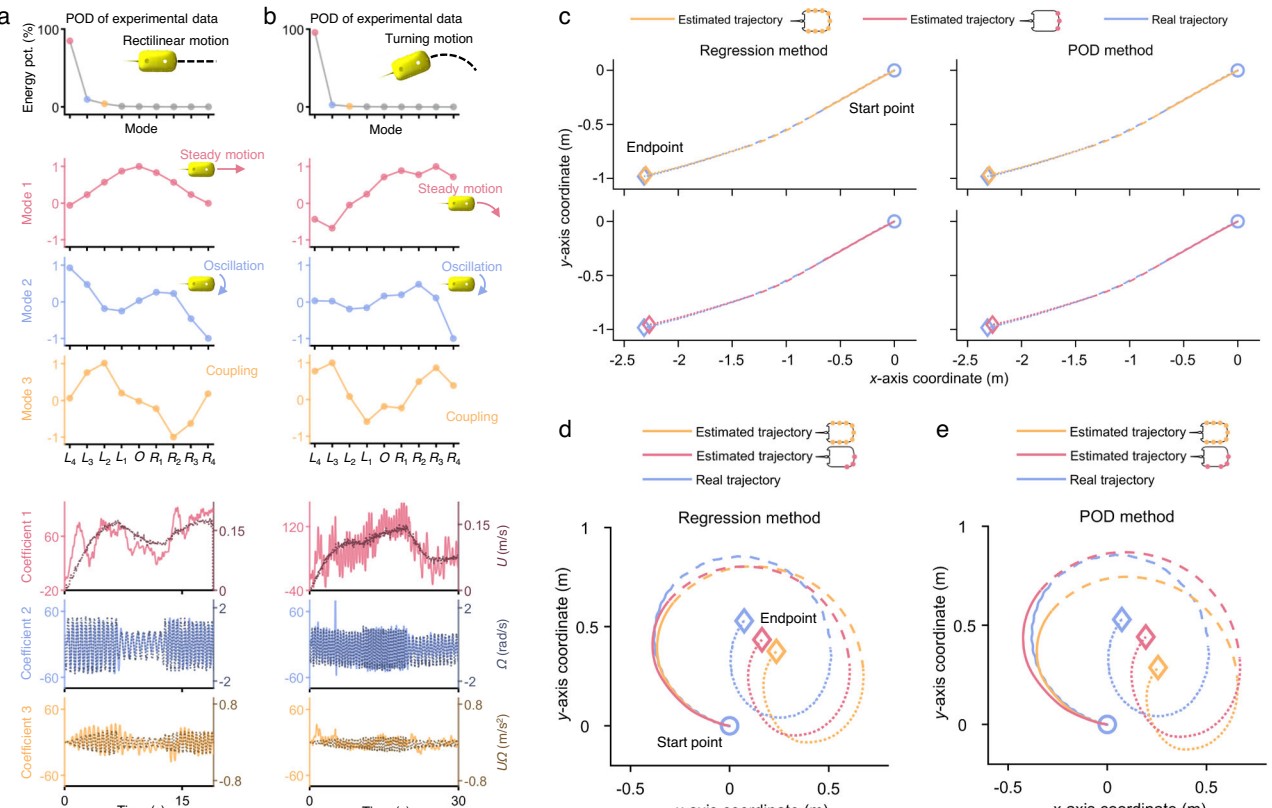

**Fig. 6 | Generalizability verification under varying and dynamic oscillation parameters.** In the rectilinear motion (**a**, **c**), the oscillation parameters are frequency = 1.8 Hz, amplitude = 30°, offset = 0°, then frequency = 1.4 Hz, amplitude = 20°, offset = 0°, and finally frequency = 2 Hz, amplitude = 25°, offset = 0°. In the turning motion (**b**, **d**, **e**), the oscillation parameters are frequency = 1.8 Hz, amplitude = 20°, offset = 25°, then frequency = 2 Hz, amplitude = 20°, offset = 20°, and finally frequency = 1.4 Hz, amplitude = 20°, offset = 30°. **a**, **b** The energy percentage and the dominant modes are similar to the situations where the oscillation

parameters are fixed. Coefficient 1 (red solid lines) reflects the variations of forward velocity (red dashed lines). Coefficient 2 (blue solid lines) reflects the variations of oscillation amplitude and frequency (blue dashed lines). **c**–**e** The trajectory estimation is improved by using the optimal combination of pressure sensors (red lines), compared with using all nine sensors (orange lines) (detailed in Supplementary Movies 5–8). The trajectories of the three different phases are represented by solid lines, dashed lines, and dotted lines, respectively.

Secondly, we verify the applicability of POD for decomposing three-dimensional pressure variations on the surface while the fish-like robot swims. Although Lighthill's pressure model, as presented in Eq. (3), is inherently two-dimensional, it can be extended to a three-dimensional framework by incorporating coordinates in all three dimensions. To test this, we conduct three-dimensional CFD simulations with our fish-like robot using an open-source IBAMR software [52–54] (detailed in Supplementary Note 10), which has been successfully used in simulating the fish-like swimming [55]. The three-dimensional spatio-temporal pressure variations on the surface can also be decomposed into several interpretable modes with coefficients using POD. As illustrated in Fig. 7a, consistent with previous sections, the first several modes are dominant. Mode 1 captures the pressure distribution caused by forward motion, characterized by a stagnation point on the head part and decreasing pressure along the sides with left-right symmetry. Coefficient 1 varies consistently with the swimming velocity, suggesting that the fish-like robot gradually accelerates to a constant velocity. Mode 2 displays asymmetry, with the maximum value on the left and the minimum value on the right, caused by body oscillation. And coefficient 2 reflects variations in angular velocity. Additionally, given that Lighthill's pressure model can adapt to various body morphologies with different coefficients $C_i$, we further test our method with three-dimensional pressure data of an eel-like swimmer in the laminar flow, which features different morphologies and swimming styles. This swimmer, with a predetermined morphology, is constructed from elliptical disks based on specified axis width and

height parameters, following the methodology outlined by Kern and Koumoutsakos [56] and Chao et al. [57]. The swimmer's position relative to the flow is fixed, featuring in-place lateral oscillations (detailed in Supplementary Note 10). The decomposition of pressure variations is shown in Fig. 7b. Mode 1 also has the maximum value on the head, as a stagnation point caused by the oncoming flow. The pressure along the sides is negative due to the velocity of the passing flow. Coefficient 1 remains nearly constant, reflecting the steady velocity of the oncoming flow. Mode 2 exhibits an asymmetric pressure distribution resulting from the oscillation of the fish body, and the sinusoidal variations in coefficient 2 also reflect this oscillation. Overall, the POD results for the pressure variations can be interpreted by Lighthill's pressure model, whether the data is two-dimensional or three-dimensional and regardless of the morphologies and swimming styles of the fish-like robots.

To further generalize our proposed method for self-state estimation from the perspective of applications, we conduct experiments with the fish-like robot swimming in flows with vortices shedding from a preceding robot, which is a typical complex flow environment in fish schools, and use ALLS to estimate the self-velocity (detailed in the "Methods" section). The two fish-like robots are fixed to a moving platform (Fig. 8a, b) for a simplification since currently no methods have been developed to control two fish-like robots swim freely and maintain the formation. Here, one robot (the leading) continuously generates vortices through body oscillation and tail flapping. In this way, the ALLS data of the focal (the following) fish-like robot includes

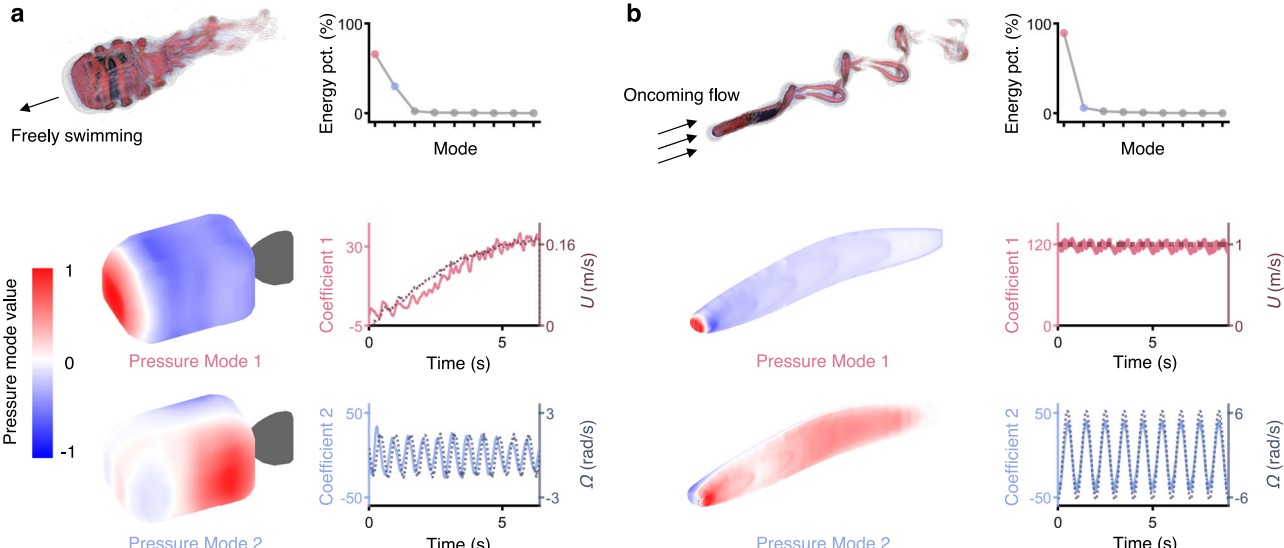

**Fig. 7 | Generalizability verification under different three-dimensional morphologies and swimming styles.** Mode decomposition (POD) of the three-dimensional pressure data from CFD simulations for numerical boxfish and eel-like models, including the flow field visualized by isosurfaces of the Q-criterion, energy proportions, modes and coefficients. **a** The boxfish model follows the same kinematics as the experiment with oscillation frequency = 1.8 Hz, amplitude = 30°, offset = 0°. **b** The eel-like model swims in the laminar flow of 1 body length per second. Modes 1, corresponding to the pressure caused by steady motion, have the largest value at the head as the stagnation point. Modes 2, corresponding to the pressure caused by oscillation, exhibit left-right antisymmetric distribution. Coefficients 1 reflect the relative velocity with respect to the flow, while coefficients 2 reflect the body's oscillation.

interference from vortices generated by the neighboring robot, which has a negative effect on estimating the self-velocity. Comparing the decomposition of pressure from different experimental platforms, it can be seen that the modes of the vortex generator are similar to those of the free-swimming fish-like robot (Fig. 8c), indicating that the setup here is equivalent to free swimming. The modes of the focal fish-like robot are slightly different because of the interference. Coefficient 1 reflects that the fish-like robots accelerate from rest to a constant velocity. For two towed robots, large fluctuations at the beginning are caused by the sudden acceleration of the moving platform. Coefficient 2 reflects the oscillation. These interpretations are also consistent with the above sections and demonstrate the rationality of the experimental platform here. As shown in Fig. 8d, Supplementary Fig. 22 and Supplementary Note 11, with the interference, the errors of self-velocity estimation are larger when the longitudinal distance is less than 30 cm. In these situations, the sensors of the focal fish-like robot are negatively affected by the vortices shedding from the leading robot. When the longitudinal distance is greater than 30 cm, the errors decrease due to the dissipation of vortices. As for the comparison between the two methods, the errors are decreased by 42.95% on average for the focal fish-like robot under interference using the POD method. Our proposed method based on POD, which can extract the main components and reduce the impact of noise, is more robust than the regression method in the presence of interference. To explain this, a rough model for error analysis is established (detailed in Supplementary Note 11).

## Discussion

In this work, we apply POD to estimate the self-velocity and trajectory for a fish-like robot based on ALLS. The decomposed modes are interpretable through Lighthill's theoretical pressure model, which indicates their generalizability. Moreover, the decomposition method also provides the specific number and locations of sensors for accurate trajectory estimation, which is verified by synthetic analysis and explained by flow visualization. Finally, we demonstrate the generalizability of this method from three perspectives: (1) trajectory estimation of the fish-like robot under varying oscillation parameters; (2) decomposition of three-dimensional pressure variations for robots with different morphologies and swimming styles; (3) robust self-velocity estimation in complex flows with vortices.

Compared to our previous studies in self-state estimation based on ALLS, our current method improves in several ways: (1) Instead of just focusing on simple correlations between the pressure data and motion states[16,36], mode decomposition method can provide more information about the components and is interpretable through Lighthill's pressure model, which aids in understanding the efficacy of ALLS; (2) The mode decomposition method could also help optimize the number and locations of ALLS sensors for self-state estimation, reducing redundancy; (3) The method is generalizable for different situations, and also robust in estimating self-states under interference since POD can extract the main components of ALLS data[43], eliminating noise and redundant information to improve the accuracy and reliability[58,59].

POD also offers a straightforward method to approximate the sensitivity of ALLS sensors to optimize their number and arrangement, rather than evaluating every possible scenario for free-swimming fish-like robots. Most previous studies in ALLS optimization are with stationary physical models[39,41,42,60,61] without considering the effect of oscillatory motion on the perception. In addition, the optimization process primarily fits specific goals[40], which lacks intuitiveness, interpretability, and generalizability. POD suggests that the spatiotemporal pressure caused by the swimming of the fish-like robot can be reduced to a lower dimension. Consequently, our ALLS design, which incorporates nine sensors, is redundant for accurately estimating the self-states. There is typically a threshold beyond which adding more sensors yields diminishing returns[40,61]. Initially, additional sensors can significantly enhance performance by providing valuable information. However, once this threshold is surpassed, each additional sensor may provide redundant information while introducing more noise, failing to contribute meaningful insights and potentially undermining the overall performance of the system.

Recently, neural networks have been explored for flow estimation and object identification using ALLS[19,27,32,38]. While these methods can achieve good fitting performance in a specific problem given sufficient

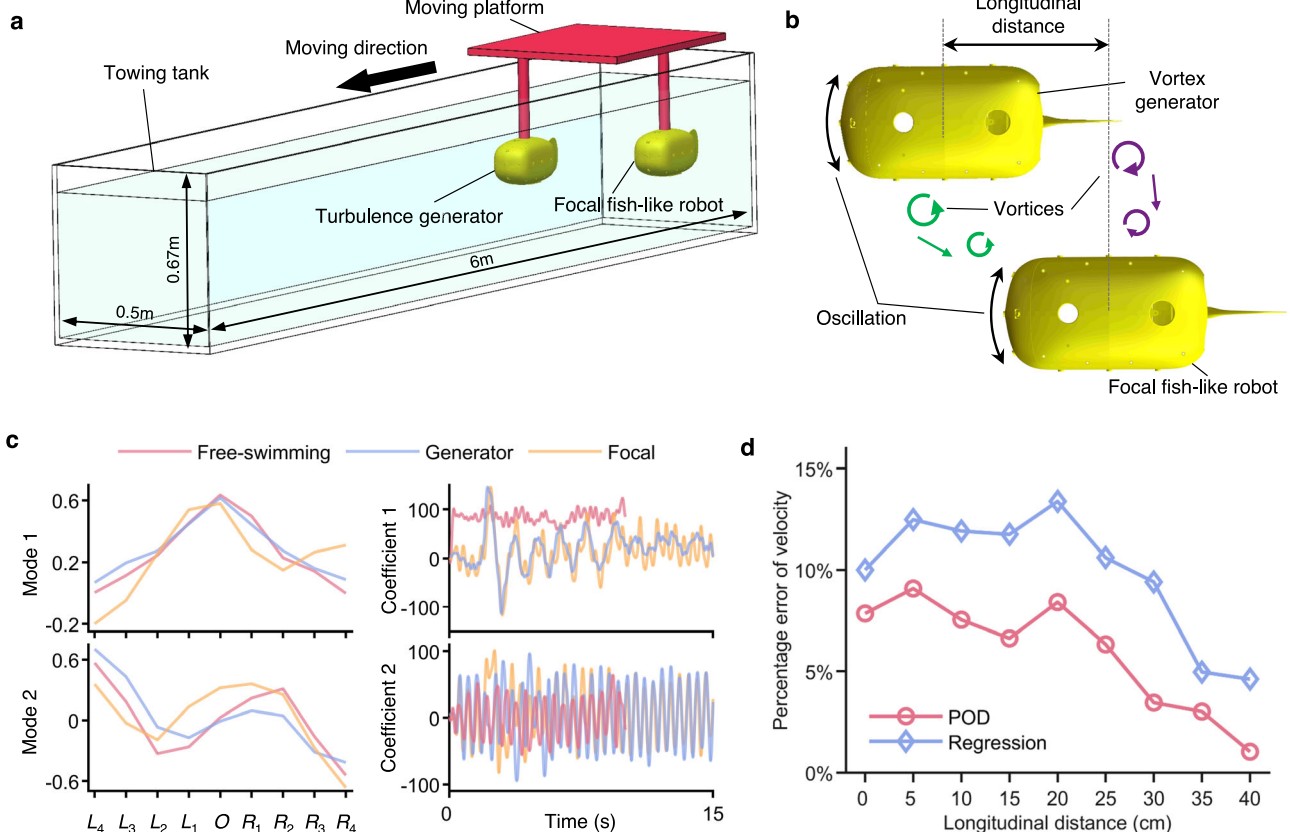

**Fig. 8 | Generalizability verification with self-velocity estimation in complex flows with vortices shedding from a neighboring robot. a** The experimental platform includes a towing tank and a moving platform whose velocity can be adjusted. Two fish-like robots are towed forward by the moving platform (detailed in Supplementary Movie 9). **b** The leading fish-like robot is used to generate vortices through body oscillation and tail flapping. The following fish-like robot uses ALLS to estimate the self-velocity with interference from vortices. The longitudinal distance between the two robots along the swimming direction can be adjusted. **c** POD of pressure data from different experimental platforms. Red, blue, and orange lines represent a free-swimming fish-like robot, the vortex generator, and the focal fish-like robot here. **d** Errors of the estimated velocity for the focal fish-like robot. The error is quantified as a percentage relative to the forward velocity. Red lines and circles represent the average errors by the POD method. Blue lines and diamonds represent the average errors by the regression method.

training data, they often require large and high-quality datasets, extensive computational resources, and may lack interpretability and generalizability. In contrast, POD leverages mode decomposition techniques to extract meaningful physical features from the pressure signals, reducing data dependency while maintaining robust performance. This not only enhances the efficiency and generalizability of our method but also provides deeper insights into the hydrodynamic sensing mechanisms that could be applicable to both robotic and biological systems. We choose POD rather than other mode decomposition methods, such as dynamic mode decomposition (DMD), to analyze the pressure data because POD is the simplest and can process time-varying systems (fish-like robot accelerating from rest) through projection and provide accurate results for further estimation. Other mode decomposition methods could be better in analyzing the evolution of the dynamic system[45,46], which could offer a different perspective on perception issues.

The interpretability and generalizability of this method for fish-like robots with different morphologies and swimming styles also show promise in analyzing how real fish use lateral lines to estimate their states, where the lateral lines involve a larger number of neuromasts to collect fluid information[62]. For instance, using the existing RoboTwin platforms[63], we can simulate the detailed flow field around real fish by using fish-like robots that share the same morphologies and kinematics. With the pressure and velocity information, our method could identify which regions of the neuromasts may play a dominant role in hydrodynamic perception. This method could further enhance

our understanding of the mapping between flow field sensing and self-states in both artificial and biological systems[64–66].

POD remains effective across the entire range of operational parameters for the fish-like robot and maybe for real fish as well. However, this method may struggle to extract meaningful pressure modes at those extreme cases, such as very low swimming velocity due to the increased influence of viscous forces[67] and the weakening of effective pressure signals, or extremely high velocity due to cavitation effects[68]. All these would be our main future work to extend our method to handle these extreme and complex cases. Moreover, POD has also demonstrated its potential for decomposing velocity fields (detailed in Supplementary Note 12). We also plan to integrate additional shear stress or velocity sensors on the surface, as real fish do, to better accommodate natural flow conditions and enhance both data quality and diversity for POD analysis. We believe this further enhancement will bring the robot's sensory capabilities closer to those of real fish, going beyond only depending on pressure signals and estimating self-states in 2D motions[69].

## Methods
### Fish-like robot
The fish-like robot (Fig. 1a) is inspired by a kind of boxfish (*Ostracion cubicum*) in nature. The size (length × width × height) is approximately 29.1 × 11.6 × 13.3 cm. The IMU (MPU6050 from TDK) combines a three-axis gyroscope and a three-axis accelerometer. So the three-dimensional acceleration and angular velocity of the fish-like robot

can be measured. The ALLS comprises 11 pressure sensors (MS5803-01BA from TE Connectivity) on the surface of the fish-like robot.

## Experiment with a free-swimming fish-like robot

The experiments are carried out in a tank, whose size (length × width × height) is $3 \times 2 \times 0.8$ m (Fig. 1b). The depth of water is 0.5 m. For the rectilinear motion, the amplitude of tail oscillation ranges from 5° to 30° with an interval of 5°. The frequency ranges from 1 Hz to 2 Hz with an interval of 0.2 Hz. The offset is fixed at 0°. For the turning motion, the amplitude is fixed at 20°. The frequency ranges from 1 Hz to 2 Hz with an interval of 0.1 Hz. The offset ranges from 20° to 40° with an interval of 5°. For each combination of parameters, the experiments are repeated five times. The sampling frequency of the IMU and ALLS data is 50 Hz. Both the IMU and ALLS data are processed by a low-pass filter. Since the value measured by the ALLS sensor is absolute pressure, thus in each experiment, we calibrate each sensor by putting the fish-like robot stationary in the still water for 10 seconds and measuring the average pressure as a reference value. While the fish-like robot is swimming, what we truly care about is the pressure variations, which is the difference between the measured pressure and the reference value.

## Methods for trajectory estimation

The yaw angle $\tilde{\theta}_i$ is integrated from IMU data and the velocity $\tilde{U}_i$ is estimated by ALLS data, respectively. The trajectory of the fish-like robot can be estimated by

$$\begin{aligned} \tilde{x}_i &= \tilde{x}_{i-1} + \tilde{U}_i \cos \tilde{\theta}_i \Delta t \\ \tilde{y}_i &= \tilde{y}_{i-1} + \tilde{U}_i \sin \tilde{\theta}_i \Delta t \end{aligned} \quad (4)$$

Concerning the regression method, according to Bernoulli's equation, the hydrodynamic pressure variation is proportional to the square of velocity. So the velocity is estimated by

$$\tilde{U} = \min_U \left\| \overline{\mathbf{p}_{\text{test}}(t)} - (\mathbf{a}U^2 + \mathbf{b}U + \mathbf{c}) \right\|^2 \quad (5)$$

where $\overline{\mathbf{p}_{\text{test}}(t)} \in \mathbb{R}^N$ represents a column vector whose element is the average hydrodynamic pressure variation at a point on the surface in the past one second (50 samples) in the test case and $\mathbf{a}$, $\mathbf{b}$, $\mathbf{c} \in \mathbb{R}^N$ represent the regression coefficients. In the training process, the regression model for each sensor is trained separately. Considering that the fish-like robot accelerates from stationary, we only select the data segments with stable velocity for training. The pressure and swimming velocity are averaged in these segments and each trial generates only one pair of data points for training.

Concerning our proposed method based on POD, the relationship between the coefficient of mode 1 and the velocity can be roughly described by a quadratic regression model (Supplementary Fig. 4). The velocity is estimated by

$$\tilde{U} = \min_U \left| \overline{\text{coef}_{\text{test}}(t)} - \left( aU^2 + bU + c \right) \right|^2 \quad (6)$$

where $\overline{\text{coef}_{\text{test}}(t)}$ represents the average value of the coefficient of mode 1 in the test case and $a$, $b$, $c$ represent the regression coefficients. The coefficient is also calculated using the data of the past one second (50 samples).

## Analysis on trajectory estimation

Nine points on the surface of the fish-like robot where pressure sensors are installed are alternative points. We analyze the estimation errors of all arbitrary combinations of $n(n = 1, .., 9)$ sensors and verify whether the error is minimized by optimally selecting the number and locations of sensors according to POD. For the turning motion, in the case of

using $n$ sensors, the trajectory estimation errors of $C_9^n$ combinations are calculated. In order to reduce the impact of accidental deviations, we define the error contribution for each sensor as follows.

$$e_i = \frac{1}{C_8^{n-1}} \sum_{i \in c_k} \text{err}_{c_k}, \, i = 1, \ldots, 9 \quad (7)$$

where $\text{err}_{c_k}$ represents the average error using the sensor combination $c_k$. $c_k$ is a set of serial numbers representing the sensors used in the case. The error contribution means the average performance of all sensor combinations which include the $i$th sensor. The total number of such combinations is $C_8^{n-1}$. Instead of directly selecting the sensor combination of the minimum error, the sensors with lower error contribution are selected to form an optimal combination. It should be avoided that some sensors at bad locations perform better than those at good locations due to accidental deviations in experiments. Furthermore, after finding the optimal combination for each number of sensors $n(n = 1, .., 9)$, we analyze the changes in the trajectory estimation errors with the number of used sensors $n(n = 1, .., 9)$ to investigate the redundancy of ALLS. For the rectilinear motion, considering the symmetry of the body, the analysis can be simplified to a certain extent.

## Hydrogen bubble flow visualization

The size (length × width × height) of the test section in the water channel is $1 \times 0.4 \times 0.5$ m (Supplementary Fig. 19a). Considering that it is difficult to follow and observe a free-swimming fish-like robot, we fix a 3D printed model of the same size in the uniform flow. The model is hollowed from the right side and hung on the bracket installed on the water channel. The bracket consists of three parts, sticks AB, AD, and BC (Supplementary Fig. 19b), which are assembled with the model by an electronic motor (point A) and three rotary hinges (points B, C, D). Point D is located at the rotation center of the fish-like robot during swimming. The motor rotates stick AB according to a sinusoidal function and drives the model simultaneously. The oscillation of the fish body is set to be the same as that of a free-swimming fish-like robot (detailed in Supplementary Note 9 and Supplementary Fig. 20) and the flow velocity is equal to the swimming velocity for a better simulation. The voltage of the water electrolysis is 60 V. Hydrogen bubbles are generated at the frequency of 50 Hz with a duty cycle of 0.2. The images are captured by a high-speed camera at the rate of 300 fps. It should be noted that the model is rotated by 90° compared with the free-swimming fish-like robot. Since the flow in the recirculating water channel is uniform in three-dimensional space, the two-dimensional flow structures around the fish body can be simulated and observed in our flow visualization experiment.

## Experiment in complex flows with vortices

The size (length × width × height) of the tank is $6 \times 0.5 \times 0.67$ m. Two fish-like robots are fixed to a moving platform (Fig. 8a). The forward displacement of the fish-like robots is entirely controlled by the moving platform. The platform is above the tank and moves straight at a given velocity of 0.122 m/s. The tail oscillation parameters of two fish-like robots are frequency = 2 Hz, amplitude = 25°, and offset = 0°. Considering the relative width of the tank and two fish-like robots, the lateral distance is fixed at 16 cm, which ensures that the ALLS sensors perform well[70]. If the lateral distance is too small, it may cause collisions between the fish-like robots, and if it is too far, it brings the fish-like robots close to the walls, where the ALLS sensors are influenced negatively by the wall effect. The longitudinal distance varies from 0 to 40 cm (1.5 body length) at an interval of 5 cm. Due to the constraints on the motion, we only use ALLS data to estimate the swimming velocity of the focal fish-like robot under the interference generated by the neighboring robot, which is compared with the velocity of the moving

platform. Considering that the specific velocity is unknown in the acceleration process of the moving platform at the beginning and the deceleration process at last, we only select the data segments between these two stages for analysis. It is worth mentioning that the estimation model is trained only using the data from free-swimming experiments and then directly used to estimate the self-velocity of the focal fish-like robot here.

## Data availability
The data that support the findings of this study are available in figshare with the identifier https://figshare.com/s/8c0b1e09cc3e75836f6c. Source data are provided with this paper.

## Code availability
All the data analyses were performed using custom scripts written in MATLAB R2021a. All codes that support the findings of this study are available in figshare with the identifier https://figshare.com/s/8c0b1e09cc3e75836f6c.

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

## Acknowledgements

G.X., Y.Z., S.L., and M.X. are supported in part by the National Natural Science Foundation of China under Grant U22A2062, Grant 12272008, and Grant U23B2037 and in part by the Beijing Natural Science Foundation under Grant 3242003. X.Z. gratefully acknowledges funding from the Natural Science Foundation of Zhejiang Province (Grant No. LZ25F030002). L.L. and L.C. are supported by the Max-Planck Society, the Deutsche Forschungsgemeinschaft (DFG, German Research Foundation) under Germany's Excellence Strategy–EXC 2117-422037984 and Messmer Foundation Research Award. G.X. and L.L. are supported by the Sino-German Centre in Beijing for generous funding of the Sino-German mobility grant M-0541. Y.J. is supported by the National Natural Science Foundation of China through Grant 92252204, Grant 12388101.

## Author contributions

Y.Z., X.Z., L.L., and G.X. conceived the idea and designed the project; Y.Z., X.Z., L.C., S.L., M.X., and Y.J. conducted the experiments and collected the data; Y.Z. and X.Z. analyzed the data; Y.Z., X.Z., and L.L. wrote the initial draft of the manuscript and all authors revised the manuscript.

## Funding

## Competing interests

The authors declare no competing interests.
