## [Transparent Peer Review file · Nature Communications]

An Interpretable Approach to Estimate the Self-motion in Fish-like Robots Using Mode Decomposition Analysis

Corresponding Author: Professor Guangming Xie

Version 0:

Reviewer comments:

Reviewer #1

(Remarks to the Author)

General comments:

I enjoyed reading this article. It addresses a difficult problem, namely how to estimate flow characteristics around a fish-like robot. I liked the fact that it took an interpretable approach rather than a black-box machine learning approach. That provides more interesting insights into the physics of fish-like locomotion.

The study seems to be carefully done, with good experimental set-ups. To increase the impact of the article I suggest the following:

I would like to see a deeper study of the effect of different frequencies and amplitudes on the decomposition method and the flow estimations. How good are the speed estimations for different frequencies? And does the method work when the frequencies and amplitudes are changing? (i.e. for non-steady state conditions, e.g. an upward or downward frequency ramp for accelerations or decelerations). I think it would be important to test and report this.

In the discussion, I would have liked the authors to come back to their observation that adding more sensors leads to worse results. I agree with the authors that this is surprising, and I think it deserves a deeper discussion.

It would be useful and important to add videos of the robot swimming in the pool (single + two-robot mode), ideally with a data traces of actual and estimated forward and rotational speeds.

Specific comments :

30 « by a closing neighboring robot » “closing” sounds strange to me. I suggest rephrasing.

92 “It can swim forward and turn freely within a two-dimensional plane by adjusting the oscillating amplitude, frequency, and offset of the soft tail.” What is the buoyancy of the robot? (positive, neutral, or negative?). It would be good to provide that information here, and explain how the robot stays in a 2d plane (I guess it is positively buoyant and swims just below the surface?)

109 “represents the snapshot of N points at time t_i .” What is the time step, and how important is the choice of a particular time step?

112 “where x represents the spatial location” In which coordinate frame? (not clear here) please explain.

Figure 2: the caption is a bit too short and does not describe all quantities shown in the figure (for instance for panel C).

119 “The parameters of tail oscillation in the example are frequency = 2 Hz, amplitude = 30°, and offset = 0°.” The methods section mentions that the experiments were done with swimming at different frequencies. How good are the speed estimations for those different frequencies? And does the method work for non-steady state conditions, e.g. when the frequencies and amplitudes are changing? (e.g. an upward or downward frequency ramp for accelerations or

decelerations). I think it would important to test and report this.

152 "which indicates that the fishlike robot is rotating to the right and squeezing the water. " Squeezing does not sound very rigorous, since water is more or less incompressible. I suggest rephrasing.

164 "Furthermore, it is evident that coefficient of mode 1 and the swimming velocity vary consistently over time" why is the speed varying? I guess because the robot has not yet reached steady-state? Please discuss briefly.

199 "Due to the different flow structures at different locations on the surface, some of the sensors may have a negative impact on perception" I would expect some frequency and amplitude effects here, e.g. that some sensors might provide useful info at low frequency swimming but not at high frequency swimming, and vice-versa. As mentioned above it would be good to discuss frequency dependencies.

Figure 6d: explain longitudinal distance in the caption. Also it would be useful to report the average error as a percentage of the swimming speed (not only in absolute values).

(Remarks on code availability)

Reviewer #2

(Remarks to the Author)

This manuscript deals with a methodology to obtain information about a fishlike robot evolving in an aqueous environment with an artificial lateral line system (ALLS).

The premises is the well-established fact that many aquatic animals (incl fish and frogs) possess a LLS that allow them to access specific information about their movement and position within that environment.

This work uses experimental data from a fishlike robot, and the data is processed by a model, which is at the core of this work.

Overall, the manuscript is poorly written (e.g. "which has become a great source of inspiration for robots"), and several statements are more than questionable (e.g. "The bio-inspired artificial lateral line system (ALLS) emerges as a novel sensing system for autonomous underwater vehicles"). Work on the ALLS has started over 15 years ago, and it is not emerging now. The team by M. Triantafyllou at MIT has accomplished a lot in this area, both in terms of hardware, but also in terms of data processing and modeling. None of these studies are referenced.

Furthermore, the authors use a number of terms, which I have a hard time fully understanding. What do they mean by "Generalizable"? In what sense? Same issue with "Robust" and "Robustness". And also with "interpretable".

I found interesting the idea to construct the model based on Lighthill's theory. However, a serious limitation of that is that it is limited to two dimensions. I doubt that a full and complete estimation of the state of the robot can be obtained from a purely 2D model.

Another critical limitation is the fact that only the pressure signal is considered. In Nature, the dual nature of the sensory mechanism of the LLS gives access to two complementary signals: pressure and acceleration. I am highly skeptical that working solely on the pressure, one can obtain sufficient information to carry out a robust self-state estimation.

Lastly, the generalization with a turbulent flow puzzles me. How can the data from so few sensors be sufficient to reconstruct the effects of potentially small eddies?

For all these reasons, I cannot recommend a revision of this manuscript.

(Remarks on code availability)

Reviewer #3

(Remarks to the Author)

This work introduces a Proper Orthogonal Decomposition (POD) method for self-state estimation of a fish robot using hydrodynamic pressure. The surface pressure measurements of a fish robot performing rectilinear and turning motions with different kinematic parameters in experiments were recorded. The dominant POD modes of the experimental data are compared with the theoretical data based on Lighthill's theory, revealing the relationships between the robot velocities and the POD coefficients. The quadratic relationships between the robot's forward velocity and the POD mode 1 coefficient are derived and applied for trajectory estimation. The number and placements of pressure sensors can be optimized for

trajectory estimation based on the sensitivity of POD mode 1 and the estimation errors, and evaluated through flow visualization around the robot's body. To demonstrate the generalizability of the proposed method for velocity estimation, two robotic fish fixed to a moving platform in a staggered pattern with different longitudinal distances were conducted. The forward velocity of the focal robot is estimated using both the POD and regression methods, with the POD method proving more robust than regression under turbulence conditions.

The proposed method has great potential for practical applications and shows the possibility of using hydrodynamic sensing for the localization of an underwater vehicle. I recommend the acceptance of this paper with a few minor comments for consideration:

1. In Fig 2b, how are the dashed lines in the right panels determined? Are the mode values of POD connected to the coefficients C_i in Eq. (3)? In Fig. 2c, why is coefficient 3 associated with the term $U \cdot \Omega$ in Eq. (3), while coefficients 1 and 2 are linked to U and Ω rather than individual terms like U^2 , Ω^2 , or dU/dt , particularly the robot is accelerating according to the red lines? What might cause the phase shifts observed between coefficient 2 and Ω ? How are the mode values normalized, and are the coefficients in Fig. 2c also normalized?

2. Line # 373: $eq(4), U_i \cdot \cos(\theta_i) \cdot dt$

3. In Supplementary Fig. 3, are the data points sourced from the experiments detailed in the Methods section, line #358-369? Do these data points correspond to all nine sensors? Are coefficient 1 and swimming velocity averaged values over a specific time period?

4. In Supplementary Figs. 5 and 6, for the estimated trajectory using three sensors, are the POD-estimated velocities derived from the quadratic relationship with all nine sensors, or are they from the quadratic relationship using only the optimal three sensors? Additionally, the labels indicate the body-fixed coordinate system OXY. Shouldn't these trajectories be presented in the global inertial coordinate system oxy?

5. In Fig. 6, is the longitudinal distance defined as the tail-to-tail or tail-to-head distance?

(Remarks on code availability)

Include a description of the data sets and codes for the figures.

Test the codes on MATLAB 2023b with no issues.

Supplementary Fig. 6b (Turn_Trajectory_Improvement_POD.m) takes longer time to be drawn.

Reviewer #4

(Remarks to the Author)

This paper presents a method for estimating the speed and turning behavior of a fish-like robot using an artificial lateral line system (ALLS). The pressure data collected by the ALLS is analyzed using Proper Orthogonal Decomposition (POD) to extract dominant flow modes. A quadratic regression model is then employed to establish a quantitative relationship between these modes and the swimming velocity. The study integrates Lighthill's theoretical framework with numerical simulations (panel code) and experimental data to interpret the physical meaning of the POD analysis and enhance the understanding of self-state estimation in bio-inspired robots. Overall, this is an interesting paper, but I think this work is not ready for publication on NC, and there are some concerns that need to be addressed:

1. There is a significant mismatch between the pressure distributions derived from Lighthill's theory and the experimental results, indicating that Lighthill's model may not fully capture the complexities of real-world swimming conditions. While the panel code results align well with Lighthill's theory, it is unclear if the panel code assumes a simplified fish body shape or matches the experimental shape. If it matches the experimental shape, what could be the cause of the discrepancies in pressure distribution trends? This difference hurts the statement of "interpretable" made by authors.
2. The description of Figures 2 and 3 lacks clarity, making it difficult to understand the rationale behind key conclusions, such as why a quadratic regression model was chosen and how Lighthill's model influenced this decision.
3. The limitations of Lighthill's theory when applied to experimental results are not explicitly acknowledged, particularly regarding the differences between experimental and theoretical pressure distributions. This weakens the overall credibility of the findings.
4. Improve the clarity of the results section, especially the explanations around Figures 2 and 3. Clearly state how the conclusions were reached and provide logical connections between observations and decisions.
5. Clarify the statement, "The pressure exhibits obvious left-right symmetry and reaches the highest value at the head," as it is unclear how this indicates the gradual acceleration of the fish-like robot. Symmetry might imply steady motion, but it does not necessarily indicate gradual acceleration.
6. It is interesting to see in fig5 that fewer sensor actually works better for turning case. Have authors ever considered actively fusing specific sensor data with respect to oscillation motion? e.g., only use the laminar flow side (switching left and right)?

(Remarks on code availability)

Version 1:

Reviewer comments:

Reviewer #1

(Remarks to the Author)

My comments have been properly addressed. The new experiments with non-steady state conditions (e.g. varying frequencies + turns) are good additions. Also the new videos are informative. I recommend accepting for publication.

(Remarks on code availability)

Reviewer #2

(Remarks to the Author)

The authors have made significant effort in addressing some of my comments.

However, they still fall short to clearly address some points:

* R2.Q4/R2/A4: "generalizable" and "other complex cases". I am sorry but I understand the meaning of the work "generalizable" but precisely, I'd like to know what kind of complex cases we're talking about. I can't accept that the method is truly general. There must be a number of limitations, which have to be spelt out.

* R2.Q5: the term "turbulence" is used in a very loose way. Again, what kind of turbulence are we talking about? Homogenous isotropic turbulence? Turbulent boundary layer? etc.

* R2.Q7: in R2.A7, the authors state that they intend to use Lighthill's theory in 3D in the future. This is NOT trivial at all. Are they referring to the Poincaré–Cosserat equations? They would need to provide more details to be convincing.

* R2.Q8: this is a very important point and concern. The authors are vaguely responding to it but no mention of it is made in the manuscript. This point should be stressed as a clear limitation of the present study, and this should appear explicitly in the manuscript.

* Finally, there have been some attempts to use brute-force machine learning techniques (specifically artificial neural networks) to use the ALLS as a object identification tool. Although the approach taken by the authors is less data-intensive, it would be worth specifying that in the manuscript.

(Remarks on code availability)

Reviewer #3

(Remarks to the Author)

The authors have adequately addressed my comments and improved the manuscript's clarity. They have also included additional results to support their statements. I have no further questions and recommend it for publication.

(Remarks on code availability)

Reviewer #4

(Remarks to the Author)

I appreciate the effort the authors have made in addressing the concerns, particularly through the additional experiments, the improved interpretability of the POD modes, and the demonstrations of sensor redundancy generalizability. These enhancements significantly strengthen the manuscript's rigor and interdisciplinary appeal.

One minor suggestion for further improvement would be to explicitly connect the POD-based ALLS approach to the biological lateral line function given the general audience nature of this journal. For instance, it would be valuable to discuss whether similar mode-like processing occurs in fish and how the observed sensor redundancy and optimal placement correspond to natural systems. Adding 1–2 sentences in the Discussion to draw this connection would further enhance the bio-inspired context of the work.

Overall, I am very pleased with the improvements, and I believe the manuscript is now nearly ready for publication.

(Remarks on code availability)

Version 2:

Reviewer comments:

Reviewer #2

(Remarks to the Author)

The authors have addressed all my comments and I am no longer opposed to its publication.

(Remarks on code availability)

Dear Editors and Reviewers:

Thank you for your time and constructive comments on our manuscript entitled 'An Interpretable and Generalizable Data-Driven Model for Robust Self-State Estimation of a Fishlike Robot' (NCOMMS-24-28282). These comments are highly valuable and extremely helpful for revising and improving our paper, as well as providing important guidance for our research. We have studied the comments carefully and made comprehensive revisions. We have conducted more than 10 extra experiments involving a fishlike robot swimming under different and dynamically varying oscillation parameters, and more than 10,000 core hours of 3D CFD simulations for 3D pressure analysis. We believe these results significantly improve the clarity and quality of the manuscript. *Specifically, the comments are in black and numbered. Our answers are in blue, and the modified texts from the manuscript are in red.*

Reviewer #1 (Remarks to the Author):

R1.Q1

General comments:

I enjoyed reading this article. It addresses a difficult problem, namely how to estimate flow characteristics around a fishlike robot. I liked the fact that it took an interpretable approach rather than a black-box machine learning approach. That provides more interesting insights into the physics of fishlike locomotion.

The study seems to be carefully done, with good experimental set-ups. To increase the impact of the article I suggest the following:

R1.A1

We sincerely thank you for the thoughtful and encouraging comments. Your positive feedback on our work is highly motivating. The following suggestions are helpful for improving our work. We have revised the paper in accordance with the comments and provided detailed point-by-point responses below.

R1.Q2

I would like to see a deeper study of the effect of different frequencies and amplitudes on the decomposition method and the flow estimations. How good are the speed estimations for different frequencies?

R1.A2

Thank you for your insightful suggestion! We agree that studying the effect of different frequencies and amplitudes on the decomposition method and flow estimations is crucial.

We have conducted additional experiments with different amplitudes and frequencies for both rectilinear and turning motions of the free-swimming fishlike robot:

For the rectilinear motion,

Frequency = 2.0 Hz, Amplitude = 30°, Offset = 0° (shown in Fig. 2),

Frequency = 2.0 Hz, Amplitude = 20°, Offset = 0° (shown in Supplementary Figs. 5, 11).

Frequency = 1.2 Hz, Amplitude = 30°, Offset = 0° (shown in Supplementary Figs. 5, 11),

Frequency = 1.4 Hz, Amplitude = 15°, Offset = 0° (shown in Supplementary Figs. 5, 12),

Frequency = 1.8 Hz, Amplitude = 10°, Offset = 0° (shown in Supplementary Figs. 5, 12),

For the turning motion,

Frequency = 2.0 Hz, Amplitude = 20°, Offset = 20° (shown in Fig. 3),

Frequency = 1.1 Hz, Amplitude = 20°, Offset = 30° (shown in Supplementary Figs. 6, 13),

Frequency = 1.3 Hz, Amplitude = 20°, Offset = 25° (shown in Supplementary Figs. 6, 13),

Frequency = 1.5 Hz, Amplitude = 20°, Offset = 35° (shown in Supplementary Figs. 6, 14),

Frequency = 1.7 Hz, Amplitude = 20°, Offset = 30° (shown in Supplementary Figs. 6, 14).

We find our proposed method works for all above cases in general. As shown in Supplementary Figs. 5 and 6, the pressure data under different oscillation parameters can be decomposed into interpretable modes and coefficients. One of the figures is also shown below for your convenience.

As previously observed, we also find that the first three modes are critical for reconstructing the pressure information, as they account for nearly all of the energy. Mode 1 exhibits left-right symmetry and reaches the highest value at the head, and coefficient 1 fluctuates around a positive value because the fishlike robot has reached a stable velocity in the selected segments. Mode 2 exhibits antisymmetry, and coefficient 2 reflects the variation in angular velocity. Mode 3 and coefficient 3 reflect the coupling and other terms in Lighthill's theory.

Additionally, our method reveals additional interpretable properties after decomposition. Firstly, the energy proportion of the first mode, which is related to the steady motion, depends on the swimming velocity. A larger swimming velocity causes a larger energy proportion of the first mode. Secondly, the variation range of coefficient 2, which is related to the oscillation, depends on the oscillation amplitude of the tail and body. A larger oscillation amplitude causes a larger variation range of coefficient 2. These are because larger swimming velocities and oscillation amplitudes generate stronger hydrodynamic pressure signals around the surface of the fishlike robot.

Supplementary Fig. 5 Mode decomposition (POD) results of the experimental hydrodynamic pressure data of more cases in the rectilinear motion. The points, lines, and colors in this figure follow the same legend as Fig. 2. The first three modes almost occupy all the energy. Mode 1 and coefficient 1 are related to the steady motion. Mode 2 and coefficient 2 are related to the oscillation. Mode 3 and coefficient 3 are related to other coupling motions. **a**, Frequency = 2 Hz, amplitude = 20°, and offset = 0°. **b**, Frequency = 1.2 Hz, amplitude = 30°, and offset = 0°. **c**, Frequency = 1.4 Hz, amplitude = 15°, and offset = 0°. **d**, Frequency = 1.8 Hz, amplitude = 10°, and offset = 0°.

Main text Line 211: We conduct 8 additional cases under different oscillation parameters, and the results show that our method can effectively decompose the pressure data into three main interpretable components (detailed in Supplementary Note 6).

Supplementary Note 6: Decomposition results of more cases under different oscillation parameters...

We also performed self-state estimations using our data-driven method, and the results once again demonstrate its effectiveness and generalizability. Detailed results are provided in Supplementary Figs. 11-14. Part of the figures are also shown below for your convenience.

Both the regression and POD methods for estimating the self-trajectory of the fishlike robot

are effective. The optimal combinations of the pressure sensors are consistent with what we found before. For the rectilinear motion, the minimum error is reached by using three sensors L_1, O, R_1 for both methods, which is consistent with the prediction based on POD. The error of the estimation increases with more sensors. For the turning motion (to the right), the optimal combinations of sensors mostly appear on the right front part of the body.

Main text Line 282: The optimal combinations of sensors in 8 additional cases with different oscillation parameters also align with our reported results, shown in Supplementary Note 8.

Supplementary Note 8: Trajectory estimation of more cases under different oscillation parameters...

Supplementary Fig. 11 Trajectory estimation of more cases in the rectilinear motion. The points, lines, and colors in this figure follow the same legend as Fig. 4 and Fig. 6. The minimum error is reached by using L_1, O, R_1 for both methods. **a b c**, Frequency = 2 Hz, amplitude = 20°, and offset = 0°. **d e f**, Frequency = 1.2 Hz, amplitude = 30°, and offset = 0°.

Supplementary Fig. 13 Trajectory estimation of more cases in the turning motion. The points, lines, and colors in this figure follow the same legend as Fig. 5 and Fig. 6. The minimum error is reached by using three or four sensors on the right front part for both methods. **a b c**, Frequency = 1.1 Hz, amplitude = 20°, and offset = 30°. **d e f**, Frequency = 1.3 Hz, amplitude = 20°, and offset = 25°.

Therefore, our algorithm can work under different frequencies or amplitudes according to the above results.

R1.Q3

And does the method work when the frequencies and amplitudes are changing? (i.e. for non-steady state conditions, e.g. an upward or downward frequency ramp for accelerations or decelerations). I think it would be important to test and report this.

R1.A3

Thank you for your insightful comment. We agree that it is important to test this method under dynamic frequencies and amplitudes as well. We added two extra examples with dynamic oscillation parameters for both rectilinear motion and turning motion. In each example, the fishlike robot has three sets of oscillation parameter combinations.

For the rectilinear motion,

Example 1 (shown in Fig. 6),

Phase 1 (7 seconds), Frequency = 1.8 Hz, Amplitude = 30° , Offset = 0° ,

Decelerate to

Phase 2 (6 seconds), Frequency = 1.4 Hz, Amplitude = 20° , Offset = 0° ,

Accelerate to

Phase 3 (6 seconds), Frequency = 2.0 Hz, Amplitude = 25° , Offset = 0° .

Example 2 (shown in Supplementary Figs. 15, 16),

Phase 1 (8 seconds), Frequency = 1.8 Hz, Amplitude = 20° , Offset = 0° ,

Accelerate to

Phase 2 (6 seconds), Frequency = 2.0 Hz, Amplitude = 25° , Offset = 0° ,

Decelerate to

Phase 3 (11 seconds), Frequency = 1.5 Hz, Amplitude = 15° , Offset = 0° .

For the rectilinear motion,

Example 1 with fixed Offset (shown in Supplementary Figs. 17, 18),

Phase 1 (10 seconds), Frequency = 1.8 Hz, Amplitude = 30° , Offset = 20° ,

Decelerate to

Phase 2 (8 seconds), Frequency = 1.5 Hz, Amplitude = 20° , Offset = 20° ,

Accelerate to

Phase 3 (10 seconds), Frequency = 2.0 Hz, Amplitude = 25° , Offset = 20° .

Example 2 with fixed Amplitude (shown in Fig. 6),

Phase 1 (10 seconds), Frequency = 1.8 Hz, Amplitude = 20° , Offset = 25° ,

Accelerate to

Phase 2 (10 seconds), Frequency = 2.0 Hz, Amplitude = 20° , Offset = 20° ,

Decelerate to

Phase 3 (12 seconds), Frequency = 1.4 Hz, Amplitude = 20° , Offset = 30° .

Again, we found that our algorithm works as well. Fig. 6 is also shown below for your convenience. Fig. 6 **a**, **b** shows that the decomposed modes are similar to those in steady cases for both rectilinear and turning motions. Coefficient 1 can reflect the variations in forward velocity, including acceleration and deceleration, and coefficient 2 can reflect the variations in oscillation amplitudes and frequencies. The POD modes and coefficients are interpreted by Lighthill's theory, regardless of the dynamics of oscillation parameters. For

the trajectory estimation, the optimal combinations of the pressure sensors are consistent with those explored with fixed frequencies and amplitudes. The estimation error can be reduced by using the optimal combination of sensors, compared with using all nine sensors.

Fig. 6 Generalizability verification under varying and dynamic oscillation parameters. In the rectilinear motion (a c), the oscillation parameters are frequency = 1.8 Hz, amplitude = 30°, offset = 0°, then frequency = 1.4 Hz, amplitude = 20°, offset = 0°, and finally frequency = 2 Hz, amplitude = 25°, offset = 0°. In the turning motion (b d e), the oscillation parameters are frequency = 1.8 Hz, amplitude = 20°, offset = 25°, then frequency = 2 Hz, amplitude = 20°, offset = 20°, and finally frequency = 1.4 Hz, amplitude = 20°, offset = 30°. a b, The energy percentage and the dominant modes are similar to the situations where the oscillation parameters are fixed. Coefficient 1 (red solid lines) reflects the variations of forward velocity (red dashed lines). Coefficient 2 (blue solid lines) reflects the variations of oscillation amplitude and frequency (blue dashed lines). c-e, The trajectory estimation is improved by using the optimal combination of pressure sensors (red lines), compared with using all nine sensors (orange lines) (detailed in Supplementary Videos 5-8). The trajectories of the three different phases are represented by solid lines, dashed lines, and dotted lines, respectively.

Main text Line 327: The generalizability of our method has been demonstrated from three aspects. Firstly, we carry out the unsteady free-swimming experiments of one fishlike robot under varying and dynamic oscillation frequencies, amplitudes, and offsets.....More unsteady cases with similar results are shown in Supplementary Note 8. Overall, our method for interpreting the decomposition and selecting the optimal combination of sensors to estimate the self-trajectory maintains its effectiveness under varying oscillation parameters.

Supplementary Note 8: Trajectory estimation of more cases under varying oscillation parameters...

R1.Q4

In the discussion, I would have liked the authors to come back to their observation that adding more sensors leads to worse results. I agree with the authors that this is surprising, and I think it deserves a deeper discussion.

R1.A4

Thanks for the suggestion. It deserves a deeper discussion that adding more sensors leads to worse results. Some related papers reach similar conclusions that performance cannot be improved anymore if the number of sensors exceeds a threshold and even more sensors cause worse performance [1][2]. We have further discussed such a conclusion and revised the Discussion part of the manuscript.

Main text Line 444: POD suggests that the spatiotemporal pressure caused by the swimming of the fishlike robot can be reduced to a lower dimension. Consequently, our ALLS design, which incorporates nine sensors, is redundant for accurately estimating the self-states. There is typically a threshold beyond which adding more sensors yields diminishing returns. Initially, additional sensors can significantly enhance performance by providing valuable information. However, once this threshold is surpassed, each additional sensor may provide redundant information while introducing more noise, failing to contribute meaningful insights and potentially undermining the overall performance of the system.

R1.Q5

It would be useful and important to add videos of the robot swimming in the pool (single + two-robot mode), ideally with a data traces of actual and estimated forward and rotational speeds.

R1.A5

We apologize that the supplementary videos for the paper are missing in the last version. In the new version, we have added several supplementary videos, including Supplementary Videos 1, 5-9.

As for one single robot swimming, we have added videos for our experiment setups (Supplementary Video 1) and examples with varying frequencies and amplitudes (Supplementary Videos 5-8) in this revision. In each video, we present the real trajectory and estimated trajectory, as well as the real velocity and estimated velocity. In addition, we also present the real yaw angle measured by the IMU, which is directly used in the estimation, instead of the rotational velocity. Our experimental setups are similar to our previous work [3].

As for the two robots swimming, we have added videos for our experiment setups (Supplementary Video 9). Due to the 6-meter length of the towing tank and the limitations

of the experimental space, it is difficult for us to capture the trajectories of the two fishlike robots using a camera from above. Moreover, we only selected the segments of the steady motion for analysis and directly compared the estimated velocity with the constant real velocity. Since the movement of the fishlike robot is relatively simple due to the constraints, we did not estimate the trajectories. Therefore, we hope the existing figures can clearly present our estimation results. Our experimental setups are similar to our previous work [4][5].

R1.Q6

Specific comments:

30 « by a closing neighboring robot » “closing” sounds strange to me. I suggest rephrasing.

R1.A6

Thank you for the feedback, and we apologize for the misleading phrasing. To clarify, we intend to express ‘another fishlike robot near this fishlike robot’. So we delete ‘closing’ and only keep ‘neighboring’.

Main text Line 33: and robustly estimating the self-velocity in turbulent environments caused by a neighboring robot.

R1.Q7

92 “It can swim forward and turn freely within a two-dimensional plane by adjusting the oscillation amplitude, frequency, and offset of the soft tail.” What is the buoyancy of the robot? (positive, neutral, or negative?). It would be good to provide that information here, and explain how the robot stays in a 2d plane (I guess it is positively buoyant and swims just below the surface?)

R1.A7

Thank you for the feedback. We apologize for not including this information earlier. The static buoyancy is close to neutral with a slight positive to ensure the robot is just below the water surface, leaving only the top antenna used for communication with the computer above the surface. It is equipped with a system inside for adjusting the center of gravity, which allows for the adjustment of its pitch and roll angles while still in the water. Before each experiment, we adjust the pitch and roll angles to be close to zero degrees, ensuring that the fishlike robot swims within the 2D plane at a consistent depth. We have updated Fig. 1b in the manuscript to include this explanation for clarity.

Main text Line 102: Its density is slightly less than one so that almost the entire body is submerged below the water surface, except the antenna used for communication with the computer. The fishlike robot has a system inside for changing the center of gravity, which is used for adjusting the pitch and roll angles to around zero before swimming. Therefore, it can swim forward and turn freely within a two-dimensional plane at a consistent depth by adjusting the oscillation amplitude, frequency, and offset of the soft tail.

R1.Q8

109 “represents the snapshot of 픽 points at time t_{pick} .” What is the time step, and how important is the choice of a particular time step?

R1.A8

Thank you for the comment. In our manuscript, the time step is 0.02s because the sampling frequency of the pressure sensors is 50Hz. We used all pressure data for the POD analysis. We agree that the choice of time step is important, as it affects the resolution of the data. We choose 50 Hz as the pressure sampling rate according to our previous studies [3] to balance the computational cost and data quality for our fishlike robot swimming with a maximum tailbeat frequency of 2 Hz.

Main text Line 128: The experimental pressure (at the sampling rate of 50Hz) in the rectilinear motion...

R1.Q9

112 “where x represents the spatial location” In which coordinate frame? (not clear here) please explain.

R1.A9

Thank you for the comment. x represents the locations of all sensors around the robot fish body with a local coordinate defined based on the robot fish body. The main text is revised as:

Main text Line 125: where x represents the spatial location of pressure sensors in the body coordinate frame...

R1.Q10

Figure 2: the caption is a bit too short and does not describe all quantities shown in the figure (for instance for panel C).

R1.A10

Thank you for your feedback. We have revised the caption of Fig. 2 together with Fig. 3 as:

Main text Line 135: (Fig. 2): **a**, The energy distribution across each mode in POD shows that the first three modes account for nearly all the energy. **b**, The first three dominant modes (solid lines), as a function of pressure sensor locations, can be interpreted as representing the pressure caused by steady motion (red dashed lines), oscillation (blue dashed lines), and various coupling motions (orange dashed lines), respectively. The mode values are normalized such that the maximum absolute value is one. **c**, The coefficients of modes (solid lines), as a function of time, are interpreted as variations in forward velocity (red dashed lines), angular velocity (blue dashed lines), and coupling terms (orange dashed lines).

Main text Line 197: (Fig. 3): **a**, The energy distribution across each mode in POD shows that the first three modes account for nearly all the energy. **b**, The first three dominant modes (solid lines), as a function of pressure sensor locations, can be interpreted as representing the pressure caused by steady motion (red dashed lines), oscillation (blue dashed lines), and various coupling motions (orange dashed lines), respectively. The mode values are normalized such that the maximum absolute value is one. **c**, The coefficients of modes (solid lines), as a function of time, are interpreted as variations in forward velocity (red dashed lines), angular velocity (blue dashed lines), and coupling terms (orange dashed lines).

R1.Q11

119 “The parameters of tail oscillation in the example are frequency = 2 Hz, amplitude = 30°, and offset = 0°.” The methods section mentions that the experiments were done with swimming at different frequencies. How good are the speed estimations for those different frequencies? And does the method work for non-steady state conditions, e.g. when the frequencies and amplitudes are changing? (e.g. an upward or downward frequency ramp for accelerations or decelerations). I think it would be important to test and report this.

R1.A11

Thank you for emphasizing this suggestion again. It is very valuable and helpful to improve our work. With additional experiments conducted at varying swimming frequencies and amplitudes (see **R1.Q2 & R1.Q3**), we confirm that the proposed algorithm performs effectively under these conditions. These results demonstrate the generalizability and robustness of our algorithm.

R1.Q12

152 “which indicates that the fishlike robot is rotating to the right and squeezing the water.” “Squeezing does not sound very rigorous, since water is more or less incompressible. I suggest rephrasing.

R1.A12

Thank you for the suggestion. We agree that ‘squeezing’ is misleading, as water is indeed incompressible. A possible revision could be:

Main text Line 172: Two local maxima are located at the left rear and right front of the body, resulting from the oscillation of the fishlike robot, with the head rotating and pushing water away.

R1.Q13

164 “Furthermore, it is evident that coefficient of mode 1 and the swimming velocity vary consistently over time” why is the speed varying? I guess because the robot has not yet reached steady-state? Please discuss briefly.

R1.A13

Thank you for the comment. The velocity varies because the robot gradually accelerates to a steady velocity. Based on the analysis of dynamic frequencies and amplitudes of robot swimming reported in **R1.A3**, our method remains effective during these dynamic processes. To avoid further misleading, the main text has been revised as follows:

Main text Line 184: Furthermore, coefficient 1 and the swimming velocity vary consistently, suggesting that the fishlike robot is dynamically converging to the stable swimming velocity (red lines in Fig. 2c).

R1.Q14

199 “Due to the different flow structures at different locations on the surface, some of the sensors may have a negative impact on perception” I would expect some frequency and amplitude effects here, e.g. that some sensors might provide useful info at low frequency swimming but not at high frequency swimming, and vice-versa. As mentioned above it would be good to discuss frequency dependencies.

R1.A14

Thank you for the valuable comment. We have added more examples about the optimal number and locations of sensors for self-trajectory estimation under different frequencies and amplitudes for the rectilinear motion, turning motion and non-steady motion. As shown in Supplementary Note 8 and Supplementary Figs. 11-14, 16, 18, the optimal combinations of the pressure sensors are located in the same regions, which do not show a clear frequency or amplitude dependency. For the rectilinear motion, the optimal sensors are L_1, O, R_1 for both methods. For the turning motion, the optimal sensors mostly appear on the right front part of the body. The figures can be seen in **R1.A2** for convenience.

Main text Line 282: The optimal combinations of sensors in 8 additional cases with different oscillation parameters also align with our reported results, shown in Supplementary Note 8.

Supplementary Note 8: Trajectory estimation of more cases under different oscillation parameters...

R1.Q15

Figure 6d: explain longitudinal distance in the caption. Also it would be useful to report the average error as a percentage of the swimming speed (not only in absolute values).

R1.A15

Thank you for the comments. ‘Longitudinal distance’ here represents the distance between two robots along the swimming direction. We have added this description to the caption of Fig .8d. And we have revised this figure to report the error as a percentage of the swimming velocity for improving clarity.

Main text Line 393: The longitudinal distance between the two robots along the swimming direction can be adjusted.

Reviewer #2 (Remarks to the Author):

R2.Q1

This manuscript deals with a methodology to obtain information about a fishlike robot evolving in an aqueous environment with an artificial lateral line system (ALLS).

The premises is the well-established fact that many aquatic animals (incl fish and frogs) possess a LLS that allow them to access specific information about their movement and position within that environment.

This work uses experimental data from a fishlike robot, and the data is processed by a model, which is at the core of this work.

R2.A1

Thank you very much for reviewing this manuscript. We sincerely apologize for any confusion caused by some of the expressions in our manuscript. **The main contribution of the work is the new data-driven method which could work well for those free-swimming fishlike robots. And we systematically demonstrate the interpretability following Lighthill's theory, generalizability by the 3D boxfish model and eel-like model, and robustness under turbulence.** All these show the power of the new method and shed new light on future onboard-pressure-based swimming state estimation. Below are the revisions and responses based on your valuable comments. We hope these revisions meet your expectations and address your concerns. We look forward to your approval.

R2.Q2

Overall, the manuscript is poorly written (e.g. "which has become a great source of inspiration for robots"), and several statements are more than questionable (e.g. "The bio-inspired artificial lateral line system (ALLS) emerges as a novel sensing system for autonomous underwater vehicles"). Work on the ALLS has started over 15 years ago, and it is not emerging now.

R2.A2

Thank you for your valuable feedback. We apologize for the misleading phrasing in the manuscript and agree that work on the ALLS started over 15 years ago.

The concept of the artificial lateral line (ALL) sensor was first proposed in 2002 [6]. Since then, various types of ALL sensors have been designed and fabricated, based on different sensing principles, to measure underwater flow velocity and pressure [7-9]. These sensors have also been integrated into autonomous underwater vehicles (AUVs) to detect incoming flows [10][11], vortices [12][13], dipole sources [14][15], and obstacles [16][17]. Additionally, some feedback control tasks can be effectively implemented using ALL sensors [18][19]. However, most of the AUVs or fishlike robots are fixed to a bracket and remain static in a flow tank, rather than swim freely and perceive the environment like a real fish. Only a few studies have explored the use of ALL sensors for flow field perception with free-swimming

underwater robots [3][20]. Compared to fixed simple models, freely swimming fishlike robots more closely mimic the hydrodynamic interactions and behaviors of real fish, offering higher research significance and value. Additionally, the movement of freely swimming fishlike robots is more complicated and unstable, which demands higher robustness in perception or control methods. Finally, freely swimming robots are capable of carrying various equipment for marine development and exploration, providing greater application value. From the perspective of biomimetics, there remains a significant gap between the perception capabilities of ALL sensors and the sensory performance of real fish lateral lines. Therefore, our work aims to further investigate how a free-swimming fishlike robot can use ALL sensors for perception, especially for estimating their own states.

We have revised our statement to reflect the ongoing, long-standing influence of biological systems on robotic design.

Main text Line 22: The bio-inspired artificial lateral line system (ALLS) has attracted growing attention as a cutting-edge sensing technology for autonomous underwater vehicles, enabling enhanced perception of flow dynamics during underwater navigation.

Main text Line 38: Animals have evolved advanced sensory and motor capabilities that allow them to effectively navigate natural environments, and these biological strategies have inspired the development of underwater robots.

R2.Q3

The team by M. Triantafyllou at MIT has accomplished a lot in this area, both in terms of hardware, but also in terms of data processing and modeling. None of these studies are referenced.

R2.A3

Thank you for the valuable feedback. We apologize for not referencing the significant contributions of M. Triantafyllou and his team in this area. Their work has played a crucial role in the development of this field, particularly in terms of sensor fabrication and data processing. Their focus is on developing new MEMS sensors for underwater velocity and pressure sensing [8][9][15]. The sensors are installed on a fixed model for experiments. However, our focus is on utilizing onboard pressure sensors to estimate the swimming velocity and trajectory of free-swimming fishlike robots, laying the groundwork for more complex perception and control tasks. M. Triantafyllou's work has been of guiding significance to our research. In the revised manuscript, we have included references to their studies.

Reference in the manuscript

[13] Triantafyllou, M. S., Weymouth, G. D. & Miao, J. Biomimetic survival hydrodynamics and flow sensing. *Annual Review of Fluid Mechanics* 48, 1-24 (2016).

[25] Asadnia, M., Kottapalli, A. G. P., Miao, J., Warkiani, M. E., & Triantafyllou, M. S. Artificial fish skin of self-powered micro-electromechanical systems hair cells for sensing hydrodynamic flow phenomena. *Journal of the Royal Society Interface*, 12, 20150322 (2015).

R2.Q4

Furthermore, the authors use a number of terms, which I have a hard time fully understanding. What do they mean by "Generalizable"? In what sense?

R2.A4

Thank you for the feedback. By "generalizable," we mean that our method can be directly applied to other complex cases without any modifications. In the revised manuscript, we have included additional cases to emphasize this point, along with three further demonstrations. Firstly, our proposed estimation method and prediction for the optimal sensors can be used in experiments with varying and dynamic oscillation parameters, shown in Fig. 6. For your convenience, we have included the figure and its caption below.

Fig. 6 Generalizability verification under varying and dynamic oscillation parameters. In the rectilinear motion (a c), the oscillation parameters are frequency = 1.8 Hz, amplitude = 30°, offset = 0°, then frequency = 1.4 Hz, amplitude = 20°, offset = 0°, and finally frequency = 2 Hz, amplitude = 25°, offset = 0°. In the turning motion (b d e), the oscillation parameters are frequency = 1.8 Hz, amplitude = 20°, offset = 25°, then frequency = 2 Hz, amplitude = 20°, offset = 20°, and finally frequency = 1.4 Hz, amplitude = 20°, offset = 30°. a b, The energy percentage and the dominant modes are similar to the situations where the oscillation parameters are fixed. Coefficient 1 (red solid lines) reflects the variations of forward velocity (red dashed lines). Coefficient 2 (blue solid lines) reflects the variations of oscillation amplitude and frequency (blue dashed lines). c-e, The trajectory estimation is improved by using the optimal combination of pressure sensors (red lines), compared with using all nine sensors (orange lines)

(detailed in Supplementary Videos 5-8). The trajectories of the three different phases are represented by solid lines, dashed lines, and dotted lines, respectively.

Secondly, we applied our data-driven method to 3D pressure data across different morphologies and swimming styles, potentially extending beyond Lighthill's theory, which primarily focuses on 2D modeling. The results (see Fig. 7) demonstrate the generalizability of our method, as mode 1 continues to represent steady motion and mode 2 reflects oscillation. For your convenience, we have included the figure and its caption below.

Fig. 7 Generalizability verification under different three-dimensional morphologies and swimming styles. Mode decomposition (POD) of the three-dimensional pressure data from CFD simulations for numerical boxfish and eel-like models, including the flow field visualized by isosurfaces of the Q-criterion, energy proportions, modes and coefficients. **a**, The boxfish model follows the same kinematics as the experiment with oscillation frequency = 1.8 Hz, amplitude = 30°, offset = 0°. **b**, The eel-like model swims in the laminar flow of 1 body length per second. Modes 1, corresponding to the pressure caused by steady motion, have the largest value at the head as the stagnation point. Modes 2, corresponding to the pressure caused by oscillation, exhibit left-right antisymmetric distribution. Coefficients 1 reflect the relative velocity with respect to the flow, while coefficients 2 reflect the body's oscillation.

Finally, we demonstrate the generalizability and robustness of our proposed estimation method by using pressure data collected under turbulent conditions to estimate the self-velocity. The results (Fig. 8) highlight the effectiveness of our method and the superior performance of our method compared to the regression method. For your convenience, we have included the figure and its caption below.

Fig. 8 Generalizability verification with self-velocity estimation under turbulence. **a**, The experimental platform includes a towing tank and a moving platform whose velocity can be adjusted. Two fishlike robots are towed forward by the moving platform (detailed in Supplementary Video 9). **b**, The leading fishlike robot is used to generate vortices through body oscillation and tail flapping. The following fishlike robot uses ALLS to estimate the self-velocity under turbulence. The longitudinal distance between the two robots along the swimming direction can be adjusted. **c**, POD of pressure data from different experimental platforms. Red, blue, and orange lines represent a free-swimming fishlike robot, the turbulence generator, and the focal fishlike robot here. **d**, Estimated velocity of the focal fishlike robot. Red lines and circles represent the average errors by the POD method. Blue lines and diamonds represent the average errors by the regression method.

Main text Line 327: The generalizability of our method has been demonstrated from three aspects. Firstly,...Secondly,...To further...

R2.Q5

Same issue with "Robust" and "Robustness".

R2.A5

Thank you for your comment regarding the use of 'robust' and 'robustness' in our manuscript. We apologize for any confusion caused by these terms. To clarify, when we refer to 'robust,' we mean the ability of the proposed estimation method based on POD to maintain its accuracy and reliability under turbulence. As shown in Fig. 8 above, in the

experiments of two robots swimming, the following robot encounters the vortices shed by the leading robot, which are regarded as an interference with the pressure signal of the sensors on the following robot. In such situations, the following robot can still estimate its self-velocity using the POD method despite these disturbances. Our proposed POD method exhibits smaller errors compared to the traditional regression method, demonstrating its 'robustness' in the presence of the disturbance. We hope this clarification addresses your concern.

R2.Q6

And also with "interpretable".

R2.A6

Thank you for your comment regarding the use of 'interpretable.' Most data-driven methods function like a black box, making them difficult to interpret. This is not the case with our data-driven method, as it can be interpreted in alignment with the principles of Lighthill's classic theory. The modes and coefficients are calculated from the pressure data and the POD algorithm itself does not contain any information about the physical model. We connected these results to Lighthill's theoretical model and demonstrated the physical information reflected in each mode and coefficient.

Firstly, Lighthill's theory and POD decomposition exhibit consistency of spatiotemporal decoupling from the perspective of their formulations. Both the modes in POD and C_i in Lighthill's theory are solely dependent on spatial coordinates and are independent of time. Both the coefficients in POD and velocity components in Lighthill's theory are solely dependent on time.

As shown in Figs. 2 and 3, mode 1 corresponds to the pressure distribution generated by steady motion, while mode 2 corresponds to the pressure distribution resulting from oscillation. The coefficients vary consistently with the velocity and angular velocity, reflecting the intensity of these motions.

Figs. 2, 3 Mode decomposition (POD) results of the hydrodynamic pressure data in the rectilinear and turning motions. The parameters of tail oscillation in the example are frequency = 2 Hz, amplitude = 30°, and offset = 0°. The first column represents the POD results of experimental data. The second column represents the POD results of theoretical data from the panel method. a, The energy distribution across each mode in POD shows that the first three modes account for nearly all the energy. b, The first three dominant modes (solid lines), as a function of pressure sensor locations, can be interpreted as representing the pressure caused by steady motion (red dashed lines), oscillation (blue dashed lines), and various coupling motions (orange dashed lines), respectively. The mode values are normalized such that the maximum absolute value is one. c, The coefficients of modes (solid lines), as a function of time, are interpreted as variations in forward velocity (red dashed lines), angular velocity (blue dashed lines), and coupling terms (orange dashed lines).

It is similar to the Fourier expansion of a function, which expresses it as a sum of sinusoidal functions with the corresponding coefficients. Here, the sinusoidal functions represent the different frequency components, while the coefficients indicate the amplitudes and phases of these components. Another similar example is the mode decomposition of vibrations, which breaks down a complex vibrating system into a set of independent modes. The modes represent the characteristic patterns of motion that a system undergoes at its natural frequencies. The coefficients represent the amplitude of each mode, quantifying how much each mode contributes to the overall vibration of the system.

To conclude, the modes and coefficients from POD are also connected to the physical model, as shown above, becoming 'interpretable' rather than being a black box. We hope

this helps to understand our ‘interpretable’ method.

R2.Q7

I found interesting the idea to construct the model based on Lighthill's theory. However, a serious limitation of that is that it is limited to two dimensions. I doubt that a full and complete estimation of the state of the robot can be obtained from a purely 2D model.

R2.A7

Thank you for the feedback. Some work has used the 2D Lighthill's theory for explaining the perception mechanism of lateral line while fish swims [21-24] and for establishing the relationship between the pressure and velocity for fishlike robots with artificial lateral line sensors [3, 25-28]. Although the 2D Lighthill's theory can qualitatively describe the pressure variations around a swimming body, there are undoubtedly some discrepancies, shown in **R4.Q2&A2**.

In our work, Lighthill's model is only employed for interpreting our data-driven method. We select some important components in Lighthill's theory to reveal the physical meanings of the modes and coefficients in POD, shown in Figs. 2, 3. We demonstrate that the pressure data, whether in 2D or 3D, can be interpreted through the components of Lighthill's theory (As shown in **R2.A4** and Fig. 7). Our method is interpretable following Lighthill's theory in 2D, but can be further applied to the decomposition and sensing with pressure data on the 3D surface of our fishlike robot.

In the future, we plan to extend Lighthill's theory to a 3D version by incorporating velocity and angular velocity components in three-dimensional space (upward velocity, pitching and rolling angular velocities), making it better aligned with the actual experiments for our 3D robot. And we will also install more sensors in the 3D space on the robot to investigate the perception in 3D motions, such as the gliding and spiral motions.

R2.Q8

Another critical limitation is the fact that only the pressure signal is considered. In Nature, the dual nature of the sensory mechanism of the LLS gives access to two complementary signals: pressure and acceleration. I am highly skeptical that working solely on the pressure, one can obtain sufficient information to carry out a robust self-state estimation.

R2.A8

Thank you for your valuable comment. We agree that fish in nature can perceive both pressure and velocity signals through two types of neuromasts, which provide a more comprehensive understanding of their surrounding environment. However, both these two types of neuromasts are on the skin of real fish, which are in the boundary layer and affected by the flow. The velocity information perceived by the superficial neuromast close to the surface is attenuated by the boundary layer to a greater degree for low-frequency stimuli [29]. But the stimuli to the canal neuromasts, which perceive pressure information, would not be directly affected by the presence of the boundary layer or background flow [30], which is also the case for the self-state estimation task in fishlike robots. The canal

neuromast also has low-pass characteristics [31]. In addition, for ALLS, canal sensors also demonstrate significantly better noise immunity compared with superficial sensors [32]. Therefore, in this study (self-state estimation of the fishlike robot), we primarily focus on pressure information rather than combining both pressure and flow data.

We appreciate your suggestion for future studies. Both pressure and flow information would be used for tasks beyond self-state estimation, such as detecting neighbors and obstacles.

R2.Q9

Lastly, the generalization with a turbulent flow puzzles me. How can the data from so few sensors be sufficient to reconstruct the effects of potentially small eddies?

R2.A9

Thank you for the feedback. We apologize for any misleading. The results shown in Fig. 8 are a generalization of the applications of our method for self-velocity estimation. **We are not aiming to capture the eddies themselves; rather, we consider them as a form of interference in the process of estimating the self-velocity using onboard sensors.** We focus on testing the robustness of our proposed estimation method under complex flows generated by neighbors. To avoid any further misleading, we have rephrased the main texts as:

Main text Line 401: we conduct experiments with the fishlike robot swimming in a complex turbulent environment (detailed in the Methods section) and use ALLS to estimate the self-velocity

Main text Line 405: In this way, the ALLS data of the focal (the following) fishlike robot include turbulence information generated by the neighboring robot, which has a negative effect on estimating the self-velocity

R2.Q10

For all these reasons, I cannot recommend a revision of this manuscript.

R2.A10

Thank you for your time reviewing our manuscript. We have carefully considered your comments and made revisions to address the concerns raised. For instance, we have conducted numerous experiments and 3D CFD simulations to demonstrate the generalizability and robustness of our proposed method. The novelty of our paper has been clearly articulated, highlighting the critical differences compared to Triantafyllou's outstanding work. We believe the revised manuscripts have been enhanced both in quality and clarity. We would greatly appreciate your further understanding and consideration of the revised work.

Reviewer #3 (Remarks to the Author):

R3.Q1

This work introduces a Proper Orthogonal Decomposition (POD) method for self-state estimation of a fish robot using hydrodynamic pressure. The surface pressure measurements of a fish robot performing rectilinear and turning motions with different kinematic parameters in experiments were recorded. The dominant POD modes of the experimental data are compared with the theoretical data based on Lighthill's theory, revealing the relationships between the robot velocities and the POD coefficients. The quadratic relationships between the robot's forward velocity and the POD mode 1 coefficient are derived and applied for trajectory estimation. The number and placements of pressure sensors can be optimized for trajectory estimation based on the sensitivity of POD mode 1 and the estimation errors, and evaluated through flow visualization around the robot's body. To demonstrate the generalizability of the proposed method for velocity estimation, two fishlike robots fixed to a moving platform in a staggered pattern with different longitudinal distances were conducted. The forward velocity of the focal robot is estimated using both the POD and regression methods, with the POD method proving more robust than regression under turbulence conditions.

The proposed method has great potential for practical applications and shows the possibility of using hydrodynamic sensing for the localization of an underwater vehicle. I recommend the acceptance of this paper with a few minor comments for consideration:

R3.A1

Thank you very much for your valuable comments and recommendation. We sincerely appreciate your recognition of our work and its potential applications. We have carefully studied your comments and incorporated the necessary revisions to further improve the clarity and quality of the manuscript.

R3.Q2

1. In Fig 2b, how are the dashed lines in the right panels determined? Are the mode values of POD connected to the coefficients C_i in Eq. (3)? In Fig. 2c, why is coefficient 3 associated with the term $U \cdot \Omega$ in Eq. (3), while coefficients 1 and 2 are linked to U and Ω rather than individual terms like U^2 , Ω^2 , or dU/dt , particularly the robot is accelerating according to the red lines?

R3.A2

Thank you for your insightful comments. It is crucial to clearly explain how Lighthill's theory is used to interpret the modes and coefficients in POD for this problem.

With respect to the modes and C_i , they are closely connected. Firstly, both the modes in POD and C_i in Lighthill's theory are only related to spatial coordinates and are independent of time. Secondly, the specific values of C_i in Lighthill's theory can be calculated only if the shape of the fish body is analytical. But the shape of our fish body does not meet this condition. So we have adopted an alternative method based on the

panel method to calculate the pressure on the surface. By setting the forward velocity of the robot to a fixed value and setting the rotational angular velocity to zero, the pressure on the surface is $C_1 U^2$ according to Lighthill's theory. By setting the forward velocity of the robot to zero and setting the rotational angular velocity to a fixed value, the pressure on the surface is $C_3 \Omega^2$ according to Lighthill's theory. Then C_i can be calculated by normalization. In order to establish the connection between the POD modes of the experimental data and C_i from the theoretical model, we introduced the theoretical pressure data of a free-swimming fishlike robot based on the panel method of potential flow theory. As shown in Fig. 2 and Fig. 3, the POD modes of the theoretical data match C_i from Lighthill's theory well. And the POD modes of the experimental data are connected to C_i , with the POD modes of the theoretical data serving as a bridge.

Next, we discuss the relationship between the coefficients and the velocity components. Coefficient 1 primarily reflects the variation in velocity U . dU/dt could theoretically influence the hydrodynamic pressure, but their contributions are less dominant in this problem where the movement of the fishlike robot is relatively stable without sudden acceleration or deceleration. As shown in Supplementary Fig. 15 and Fig. 17, the decomposition results of the experimental data under varying oscillation parameters, coefficient 1 has always been positive, despite the presence of a deceleration phase ($dU/dt < 0$). The pressure distribution caused by forward velocity U and acceleration dU/dt should have a maximum value at the head as the stagnation point, decreasing towards the sides. Therefore, the components of acceleration could be coupled into mode 1 and coefficient 1. To conclude, coefficient 1 primarily reflects the variation in velocity U , while also incorporating some acceleration information in this problem. But our main goal is to estimate the swimming velocity and trajectory of the fishlike robot, so we have not placed much emphasis on the acceleration information. In future research, if we aim to use mode decomposition methods to explore the perception and control of the fishlike robot under abrupt disturbances which generate large acceleration, the acceleration terms dU/dt will become more important.

As for whether the dashed lines in the right panels are U or U^2 , we aim to present the qualitative conclusion intuitively that coefficient 1 reflects the variation in velocity, either increasing or decreasing simultaneously. For the quantitative relationship between the two, we have established a quadratic model $\overline{C_1(t)} = aU^2 + bU + c$ that includes both linear and quadratic terms, for subsequent estimation.

Coefficient 2 reflects the variations of rotational angular velocity Ω , rather than Ω^2 because coefficient 2 fluctuates between positive and negative values and has the same frequency as Ω . If coefficient 2 is related to Ω^2 , it should always be positive with a frequency twice that of the angular velocity Ω . However, this is not the case. In addition, there may exist a phase difference between coefficient 2 and the angular velocity because $d\Omega/dt$ may cause the pressure distribution on the surface similar to mode 2. The amplitude of angular acceleration is equal to the amplitude of angular velocity multiplied by $2\pi f$. So these two terms may couple in mode 2 and coefficient 2, resulting in a slight phase difference.

Coefficient 3 is related to $U\Omega$, which is drawn by comparing mode 3 with the coefficient C_5 ,

corresponding to $U\Omega$ in Lighthill's theory. However, this conclusion requires further investigation, as mode 3, with its relatively low energy proportion, is not significant and may be influenced by other factors. The hydrodynamic pressure caused by angular acceleration and lateral velocity may also exhibit the same antisymmetry as mode 3. It can explain why there are some differences between coefficient 3 and the variation in $U\Omega$. Here, we chose $U\Omega$ as the label on the right panel as a reference.

In this problem, we mainly focus on mode 1 and coefficient 1 for the velocity estimation. In the future, we would further investigate and interpret the remaining modes, exploring how to use them to predict more states for the fishlike robot, such as angular velocity and acceleration.

We have added related explanations to our manuscript and Supplementary Note 4.

Main text Line 149: We apply panel methods with a given velocity component U , V , or Ω , respectively, to calculate the coefficients C_i for the main components in Lighthill's theory (detailed in Supplementary Note 3). We then generate pressure data with U , V , and Ω , but this time coupling like swimming freely, and apply POD to the pressure data. We find that the three main modes extracted from POD correspond to the coefficients in Lighthill's theory (the second column in Fig. 2).

Main text Line 175: Mode 3 (orange lines in Fig. 2) reflects the coupling of steady motion and oscillation (detailed in Supplementary Note 4)....

Supplementary Note 4 How to correlate the coefficients in POD to the motion states in Lighthill's theory...

R3.Q3

What might cause the phase shifts observed between coefficient 2 and Omega?

R3.A3

Thank you for the comment. Like the pressure distribution caused by forward velocity U and acceleration dU/dt should have a maximum value at the head as the stagnation point, decreasing towards the sides, the pressure distribution caused by Ω and $d\Omega/dt$ may have a similar antisymmetric distribution, coupling in mode 2 and coefficient 2. This may cause the phase difference between coefficient 2 and the angular velocity. We have added this explanation to Supplementary Note 4.

Supplementary Note 4 Line 168: Additionally, there may be a phase difference between coefficient 2 and the angular velocity, as $d\Omega/dt$ could cause the pressure distribution on the surface similar to mode 2. The amplitude of angular acceleration is equal to the amplitude of angular velocity multiplied by $2\pi f$, so these two terms may couple in mode 2 and coefficient 2, resulting in a slight phase difference.

R3.Q4

How are the mode values normalized, and are the coefficients in Fig. 2c also normalized?

R3.A4

Thank you for the feedback. The modes are all normalized with the maximum absolute value as one. We normalized the modes to unify the left-side scale in the figure. We have added the explanation in the caption of Fig. 2 and Fig. 3.

The coefficients are not normalized so that these coefficients are directly used for training and estimation.

Main text Line 140: The mode values are normalized such that the maximum absolute value is one.

R3.Q5

2. Line # 373: eq(4), $U_i \cos(\theta_i) \Delta t$

R3.A5

We apologize for the typo. We have made the correction and added the term Δt to Eq. (4).

Main text Line 490:

$$\begin{aligned}\tilde{x}_i &= \tilde{x}_{i-1} + \tilde{U}_i \cos \tilde{\theta}_i \Delta t \\ \tilde{y}_i &= \tilde{y}_{i-1} + \tilde{U}_i \sin \tilde{\theta}_i \Delta t\end{aligned}\tag{4}$$

R3.Q6

3. In Supplementary Fig. 3, are the data points sourced from the experiments detailed in the Methods section, line #358-369? Do these data points correspond to all nine sensors? Are coefficient 1 and swimming velocity averaged values over a specific time period?

R3.A6

Thank you for the feedback. Yes, the data points are from the free-swimming experiment shown in Methods. The y-axis represents the POD coefficient 1 of the pressure data from the nine sensors. We have added the explanations to the caption of Supplementary Fig. 4.

In the process of training shown in Supplementary Fig. 4, we only select the pressure data and swimming velocity after the movement of the fishlike robot is stable. The coefficient 1 and swimming velocity are averaged in these segments and each trial generates only one pair of data points for training. In the process of estimation, at each time step, we used the pressure data of the past one second (50 samples) for POD and calculated the average value of coefficient 1 for estimation. More details are shown in Methods for trajectory estimation (Line 498 Line 505). We have also added more details for improving clarity.

Main text Line 497: The pressure and swimming velocity are averaged in these segments and each trial generates only one pair of data points for training.

Supplementary Fig. 4 Line 190: Quadratic relationships between the coefficient of mode 1 using all nine sensors and the swimming velocity.

R3.Q7

4. In Supplementary Figs. 5 and 6, for the estimated trajectory using three sensors, are the POD-estimated velocities derived from the quadratic relationship with all nine sensors, or are they from the quadratic relationship using only the optimal three sensors? Additionally, the labels indicate the body-fixed coordinate system OXY. Shouldn't these trajectories be presented in the global inertial coordinate system oxy?

R3.A7

Thank you for the feedback. The estimated velocities using three sensors are derived from the quadratic relationship from these three sensors. For each estimation using different combinations of sensors, we train the regression model separately using the selected sensors and estimate the velocity.

Yes, these trajectories should be presented in the global inertial coordinate system oxy. We apologize for the mistake and have made the corrections.

R3.Q8

5. In Fig. 6, is the longitudinal distance defined as the tail-to-tail or tail-to-head distance?

R3.A8

Thank you for the feedback. The longitudinal distance is defined as the distance between the mass centers of the two robots along the swimming direction. So it should be the tail-to-tail distance. We have added this description to the caption of Fig .8d for improving clarity.

Main text Line 394: The longitudinal distance between the two robots along the swimming direction can be adjusted.

R3.Q9

Reviewer #3 (Remarks on code availability):

Include a description of the data sets and codes for the figures.

Test the codes on MATLAB 2023b with no issues.

Supplementary Fig. 6b (Turn_Trajectory_Improvement_POD.m) takes longer time to be drawn.

R3.A9

Thank you for the feedback. The code execution time can be significantly reduced by using a more efficient algorithm for POD.

Line 250 in Turn_Trajectory_Improvement_POD.m

[U_X_1,S_X_1,V_X]=svd(G_X); is changed to [U_X_1,S_X_1,V_X]=svd(G_X,'econ');

Reviewer #4

R4.Q1

This paper presents a method for estimating the speed and turning behavior of a fishlike robot using an artificial lateral line system (ALLS). The pressure data collected by the ALLS is analyzed using Proper Orthogonal Decomposition (POD) to extract dominant flow modes. A quadratic regression model is then employed to establish a quantitative relationship between these modes and the swimming velocity. The study integrates Lighthill's theoretical framework with numerical simulations (panel code) and experimental data to interpret the physical meaning of the POD analysis and enhance the understanding of self-state estimation in bio-inspired robots. Overall, this is an interesting paper, but I think this work is not ready for publication on NC, and there are some concerns that need to be addressed:

R4.A1

Thank you for the constructive comments on our paper. We have carefully considered your feedback and made revisions to address the concerns raised. Below are our point-by-point responses.

R4.Q2

1. There is a significant mismatch between the pressure distributions derived from Lighthill's theory and the experimental results, indicating that Lighthill's theory may not fully capture the complexities of real-world swimming conditions. While the panel code results align well with Lighthill's theory, it is unclear if the panel code assumes a simplified fish body shape or matches the experimental shape. If it matches the experimental shape, what could be the cause of the discrepancies in pressure distribution trends? This difference hurts the statement of "interpretable" made by authors.

R4.A2

Thank you for pointing out the discrepancies between the pressure distributions derived from Lighthill's theory and the experimental results. It is indeed important to explore the potential mechanisms which cause such discrepancies.

Firstly, the shape of the fishlike robot in the panel code is the same as the experimental shape. We divided the 2D contour of our boxfish-shaped robot into many panels and put singularities on the panels. The velocity boundary conditions are fitted to calculate the strength of singularities. Then, we can obtain the potential function of the entire flow field and calculate the pressure on the surface of the robot based on the unsteady Bernoulli's equation.

With further experiments based on CFD simulation of a 3D free-swimming fishlike robot (Fig. 7) which consider the viscosity, we do find the discrepancies between the experiments and 3D simulations reduce, as shown in Supplementary Fig. 3. This indicates that the 3D morphology and viscosity, which are largely ignored in Lighthill's theory and the panel method, could be the main reasons causing such discrepancies.

Fig. 7 Generalizability verification under different three-dimensional morphologies and swimming styles. Mode decomposition (POD) of the three-dimensional pressure data from CFD simulations for numerical boxfish and eel-like models, including the flow field visualized by isosurfaces of the Q-criterion, energy proportions, modes and coefficients. **a**, The boxfish model follows the same kinematics as the experiment with oscillation frequency = 1.8 Hz, amplitude = 30°, offset = 0°. **b**, The eel-like model swims in the laminar flow of 1 body length per second. Modes 1, corresponding to the pressure caused by steady motion, have the largest value at the head as the stagnation point. Modes 2, corresponding to the pressure caused by oscillation, exhibit left-right antisymmetric distribution. Coefficients 1 reflect the relative velocity with respect to the flow, while coefficients 2 reflect the body's oscillation.

Supplementary Fig. 3 The decomposed modes 1 and 2 of experimental, CFD simulation and theoretical pressure data on the two-dimensional plane. **a**, Experimental data. **b**, CFD simulation data. **c**, Theoretical data.

“Interpretable” stands in contrast to most traditional data-driven methods, whose functions are black boxes, revealing correlations but not underlying mechanisms. By comparing our data-driven method with Lighthill’s theory, we found that the dominant modes and coefficients precisely correspond to the main components in Lighthill’s theory, even though the exact values differ slightly. Modes 1 from theoretical data, simulation data and experimental data all reflect the steady motion. Modes 2 reflect the oscillation. This discrepancy could be attributed to 3D morphologies and other complex fluid dynamics as

discussed before. And the importance or intensity of the mode is mainly described by the coefficients, which can be used for further quantitative estimations.

To conclude, Lighthill's theory here is only adopted to qualitatively interpret the physical meanings of the modes and coefficients from POD, which is a data-driven method. Based on the interpretation that mode 1 and the corresponding coefficient reflect the steady motion forward, we use coefficients and experimental velocities to train a quantitative model for further estimation.

We have added more details and explanations to Supplementary Note 3.

Main text Line 162: For the experimental data, we also find the modes extracted by POD qualitatively match the coefficients C_i from Lighthill's theory as well (the first column in Fig. 2), although there are some quantitative differences because the assumption of potential flow theory is different from the actual environments and the two-dimensional Lighthill's theory is different from the three-dimensional fish body (detailed in Supplementary Note 3).

Supplementary Note 3: Differences between the theoretical data from the panel method and experimental data...

R4.Q3

2. The description of Figures 2 and 3 lacks clarity, making it difficult to understand the rationale behind key conclusions, such as why a quadratic regression model was chosen and how Lighthill's theory influenced this decision.

R4.A3

Thank you for the feedback. We apologize for not explaining the results clearly. First of all, Lighthill's theory and POD decomposition reach a consensus for spatiotemporal decoupling from the perspective of formulations. Specifically, both the modes in POD and C_i in Lighthill's theory are only related to spatial coordinates and are independent of time. Both the coefficients in POD and the motion states in Lighthill's theory only depend on time. So all the work we have done in this section is to interpret the modes and coefficients in the data-driven method (POD) using the parameters in Lighthill's theory.

Besides, C_i in Lighthill's theory cannot be calculated directly because the shape of our fishlike robot is not analytical. So the panel method, which is based on the same assumption of the potential flow, is adopted as an alternative. If we set that the robot only has a constant forward velocity, the pressure on the surface is equal to $C_1 U^2$ according to Lighthill's theory. $C_3 \Omega^2$ can also be calculated by setting that the robot only has a constant angular velocity.

In order to establish the connection between POD results of the experimental data and Lighthill's theory, we introduced the theoretical pressure data of a free-swimming fishlike robot as a bridge. The theoretical pressure data is calculated by the panel method. It is obvious that the theoretical pressure data can be decomposed into several modes and coefficients by POD (the second column in Fig. 2), which match Lighthill's theory well. The modes represent the pressure distribution by only steady motion, in-place oscillation and other coupling motion. The coefficients reflect the variations in forward velocity, angular

velocity and other coupling terms. Then we came back to the POD results of the experimental data. The modes of experimental data are similar to the modes of the theoretical data, as well as C_i in Lighthill's theory, although there are also some quantitative differences because the assumption of potential flow theory is different from real environments. And the coefficients of the experimental data match the velocity and angular velocity in experiments. We hope this analytical process can help to understand our key conclusion that the POD decomposition results can be interpreted by Lighthill's model here.

Regarding the model selection for estimation, Lighthill's theory suggests a quadratic relationship between the pressure variations and the velocity. So we trained a quadratic regression between the two, also with a linear term and constant term to modify the theoretical model to better align with reality and improve the model's fit like [18].

We have added more details and explanations to our manuscript for improving the clarity of this section, including the logic connections and the specific explanations of each mode and coefficient.

Main text Line 149: We apply panel methods⁵¹ with a given velocity component U , V , or Ω , respectively, to calculate the coefficients C_i for the main components in Lighthill's theory (detailed in Supplementary Note 3). We then generate pressure data with U , V , and Ω , but this time coupling like swimming freely, and apply POD to the pressure data. We find that the three main modes extracted from POD correspond to the coefficients in Lighthill's theory (the second column in Fig. 2). Specifically, for the rectilinear motion, mode 1 aligns well with C_1 of U in Lighthill's theory, which represents the pressure resulting from the steady motion forward. And the corresponding coefficient 1 in POD varies consistently with the forward velocity. Similarly, we compare modes 2 and 3 extracted from POD with the pressure caused by angular velocity Ω and coupling term $U\Omega$. The modes show strong matches with the theoretical pressure distributions caused by the decomposed swimming patterns. And the coefficients reflect the varying velocity components. These findings suggest that POD can effectively decompose the pressure data into several main modes and coefficients, which are interpretable within the framework of Lighthill's theory.

For the experimental data, we also find the modes extracted by POD qualitatively match the coefficients C_i from Lighthill's theory as well (the first column in Fig. 2), although there are some quantitative differences because the assumption of potential flow theory is different from the actual environments and the two-dimensional Lighthill's theory is different from the three-dimensional fish body (detailed in Supplementary Note 3).

Main text Line 185: Since there is a quadratic relationship between the pressure variations and the velocity in Lighthill's theory (Eq. (3)), we train a quadratic regression model from experimental data and propose a novel method using this model to estimate the velocity of the fishlike robot (detailed in the Methods section and Supplementary Note 5).

R4.Q4

3. The limitations of Lighthill's theory when applied to experimental results are not explicitly

acknowledged, particularly regarding the differences between experimental and theoretical pressure distributions. This weakens the overall credibility of the findings.

R4.A4

Thank you once again for pointing out this key issue about the limitations of Lighthill's theory. We hope that our response to **R4.Q2** could address this comment. Please refer to **R4.A2** for more details.

R4.Q5

4. Improve the clarity of the results section, especially the explanations around Figures 2 and 3. Clearly state how the conclusions were reached and provide logical connections between observations and decisions.

R4.A5

Thank you for your helpful feedback. We notice that this comment is similar to the one raised earlier regarding the clarity of the results section (**R4.Q3**). To address this, we have made revisions and added more details about the explanations and the logical connections. For more details, please refer to **R4.A3**.

R4.Q6

5. Clarify the statement, "The pressure exhibits obvious left-right symmetry and reaches the highest value at the head," as it is unclear how this indicates the gradual acceleration of the fishlike robot. Symmetry might imply steady motion, but it does not necessarily indicate gradual acceleration.

R4.A6

Thank you for the comment. We apologize for not expressing this statement more clearly. It is right that symmetry can imply steady motion or acceleration because forward velocity and acceleration both cause a symmetric pressure distribution, with the highest value at the head. In our work, the contribution of acceleration is less dominant because the movement of the fishlike robot is relatively stable without sudden acceleration or deceleration. So, the symmetry only implies that mode 1 is caused by the forward motion, that is to say, mainly the velocity along the swimming direction, regardless of its values or variations. Then, the gradual acceleration of the fishlike robot is indicated by coefficient 1. Coefficient 1 reflects the variations in velocity, either increasing or decreasing simultaneously. We have revised this statement to make it clearer.

Main text Line 167: The pressure distribution of mode 1 exhibits apparent left-right symmetry, with the highest value at the head serving as the stagnation point. The variation of the corresponding coefficient (red lines in Fig. 2c) is similar to the swimming velocity U , showing that the fishlike robot gradually accelerates to a constant velocity.

R4.Q7

6. It is interesting to see in fig5 that fewer sensor actually works better for turning case.

Have authors ever considered actively fusing specific sensor data with respect to oscillation motion? e.g., only use the laminar flow side (switching left and right)?

R4.A7

Thank you for the comment. For the motion of turning right, as shown in Supplementary videos 3, 4, the sensors on the windward (right) side are always on the laminar flow side while the sensors on the leeward (left) side are always on the vortex flow side regardless of the oscillation direction (swinging to left or right). So we think the pressure data measured by the sensors on the windward side are always less affected by the turbulence.

Besides, if we used the sensor on the right while the fishlike robot is swinging to the right and used the sensor on the left while the fishlike robot is swinging to the left, the peak of the data would be kept and the valley would be neglected in each period. If so, the value of pressure variations used for estimation is not only related to the velocity, but also depends on the amplitude and frequency of fish body oscillation because larger amplitude causes larger pressure variations. In contrast, if fixed sensors are used, the effect of the oscillation can be reduced by averaging the pressure data over a period of time.

In the future, actively fusing specific sensors may be a good method for more complicated tasks. For example, if the fishlike robot swam straight and then turned right, the different optimal combinations of sensors could be used for estimation in different motions. If there were several fish or robots nearby, our fishlike robot could use different sensors facing different neighbors to locate them as desired.

Reference

- [1] Verma, S., Papadimitriou, C., Lüthen, N., Arampatzis, G., & Koumoutsakos, P. (2020). Optimal sensor placement for artificial swimmers. *Journal of Fluid Mechanics*, 884, A24.
- [2] Xu, D., Lv, Z., Zeng, H., Bessaih, H., & Sun, B. (2019). Sensor placement optimization in the artificial lateral line using optimal weight analysis combining feature distance and variance evaluation. *ISA transactions*, 86, 110-121.
- [3] Zheng, X., Wang, W., Xiong, M., & Xie, G. (2020). Online state estimation of a fin-actuated underwater robot using artificial lateral line system. *IEEE Transactions on robotics*, 36(2), 472-487.
- [4] Zheng, X., Wang, M., Zheng, J., Tian, R., Xiong, M., & Xie, G. (2019, November). Artificial lateral line based longitudinal separation sensing for two swimming robotic fish with leader-follower formation. In *2019 IEEE/RSJ International Conference on Intelligent Robots and Systems (IROS)* (pp. 2539-2544). IEEE.
- [5] Zheng, X., Xiong, M., & Xie, G. (2019, October). Data-driven modeling for superficial hydrodynamic pressure variations of two swimming robotic fish with leader-follower formation. In *2019 IEEE International Conference on Systems, Man and Cybernetics (SMC)* (pp. 4331-4336). IEEE.
- [6] Fan, Z., Chen, J., Zou, J., Bullen, D., Liu, C., & Delcomyn, F. (2002). Design and fabrication of artificial lateral line flow sensors. *Journal of micromechanics and microengineering*, 12(5), 655.
- [7] Abdulsadda, A. T., & Tan, X. (2012). An artificial lateral line system using IPMC sensor arrays. *International Journal of Smart and Nano Materials*, 3(3), 226-242.
- [8] Kottapalli, A. G. P., Asadnia, M., Miao, J., & Triantafyllou, M. (2014). Touch at a distance sensing: lateral-line inspired MEMS flow sensors. *Bioinspiration & biomimetics*, 9(4), 046011.
- [9] Triantafyllou, M. S., Weymouth, G. D., & Miao, J. (2016). Biomimetic survival hydrodynamics and flow sensing. *Annual Review of Fluid Mechanics*, 48(1), 1-24.
- [10] DeVries, L., Lagor, F. D., Lei, H., Tan, X., & Paley, D. A. (2015). Distributed flow estimation and closed-loop control of an underwater vehicle with a multi-modal artificial lateral line. *Bioinspiration & biomimetics*, 10(2), 025002.
- [11] Kruusmaa, M., Fiorini, P., Megill, W., De Vittorio, M., Akanyeti, O., Visentin, F., ... & Liszewski, A. (2014). Filose for svenning: A flow sensing bioinspired robot. *IEEE Robotics & Automation Magazine*, 21(3), 51-62.
- [12] Venturelli, R., Akanyeti, O., Visentin, F., Ježov, J., Chambers, L. D., Toming, G., ... & Fiorini, P. (2012). Hydrodynamic pressure sensing with an artificial lateral line in steady and unsteady flows. *Bioinspiration & biomimetics*, 7(3), 036004.
- [13] Yang, Y., Chen, J., Engel, J., Pandya, S., Chen, N., Tucker, C., ... & Liu, C. (2006). Distant touch hydrodynamic imaging with an artificial lateral line. *Proceedings of the National Academy of Sciences*, 103(50), 18891-18895.

- [14] Abdulsadda, A. T., & Tan, X. (2013). Nonlinear estimation-based dipole source localization for artificial lateral line systems. *Bioinspiration & biomimetics*, 8(2), 026005.
- [15] Asadnia, M., Kottapalli, A. G. P., Miao, J., Warkiani, M. E., & Triantafyllou, M. S. (2015). Artificial fish skin of self-powered micro-electromechanical systems hair cells for sensing hydrodynamic flow phenomena. *Journal of the Royal Society Interface*, 12(111), 20150322.
- [16] Yen, W. K., Sierra, D. M., & Guo, J. (2018). Controlling a robotic fish to swim along a wall using hydrodynamic pressure feedback. *IEEE Journal of Oceanic Engineering*, 43(2), 369-380.
- [17] Bouffanais, R., Weymouth, G. D., & Yue, D. K. (2011). Hydrodynamic object recognition using pressure sensing. *Proceedings of the Royal Society A: Mathematical, Physical and Engineering Sciences*, 467(2125), 19-38.
- [18] Salumäe, T., & Kruusmaa, M. (2013). Flow-relative control of an underwater robot. *Proceedings of the Royal Society A: Mathematical, Physical and Engineering Sciences*, 469(2153), 20120671.
- [19] Zheng, J., Zhang, T., Wang, C., Xiong, M., & Xie, G. (2021). Learning for attitude holding of a robotic fish: An end-to-end approach with sim-to-real transfer. *IEEE Transactions on Robotics*, 38(2), 1287-1303.
- [20] Zhang, Z., Zhou, C., Cheng, L., Wang, X., & Tan, M. (2023). Real-time velocity vector resolving of artificial lateral line array with fishlike motion noise suppression. *IEEE Transactions on Robotics*.
- [21] Lighthill, S. J. (1993). Estimates of pressure differences across the head of a swimming clupeid fish. *Philosophical Transactions of the Royal Society of London. Series B: Biological Sciences*, 341(1296), 129-140.
- [22] Rowe, D. M., Denton, E. J., & Batty, R. S. (1993). Head turning in herring and some other fish. *Philosophical Transactions of the Royal Society of London. Series B: Biological Sciences*, 341(1296), 141-148.
- [23] McHenry, M. J., Michel, K. B., Stewart, W., & Müller, U. K. (2010). Hydrodynamic sensing does not facilitate active drag reduction in the golden shiner (*Notemigonus crysoleucas*). *Journal of Experimental Biology*, 213(8), 1309-1319.
- [24] Franosch, J. M. P., Hagedorn, H. J., Goulet, J., Engelmann, J., & Van Hemmen, J. L. (2009). Wake tracking and the detection of vortex rings by the canal lateral line of fish. *Physical review letters*, 103(7), 078102.
- [25] Akanyeti, O., Thornycroft, P. J., Lauder, G. V., Yanagitsuru, Y. R., Peterson, A. N., & Liao, J. C. (2016). Fish optimize sensing and respiration during undulatory swimming. *Nature communications*, 7(1), 11044.
- [26] Qiu, C., Wu, Z., Wang, J., Tan, M., & Yu, J. (2023). Locating dipole source using self-propelled robotic fish with artificial lateral line system. *IEEE Transactions on Automation Science and Engineering*.
- [27] Akanyeti, O., Chambers, L. D., Ježov, J., Brown, J., Venturelli, R., Kruusmaa, M., ... & Fiorini, P. (2013). Self-motion effects on hydrodynamic pressure sensing: part I. Forward-

backward motion. *Bioinspiration & biomimetics*, 8(2), 026001.

[28] Kim, J. H., Mai, T. L., Cho, A., Heo, N., Yoon, H. K., Park, J. Y., & Byun, S. H. (2024). Establishment of a Pressure Variation Model for the State Estimation of an Underwater Vehicle. *Applied Sciences*, 14(3), 970.

[29] Windsor, S. P., & McHenry, M. J. (2009). The influence of viscous hydrodynamics on the fish lateral-line system. *Integrative and comparative biology*, 49(6), 691-701.

[30] Mogdans, J., & Bleckmann, H. (2012). Coping with flow: behavior, neurophysiology and modeling of the fish lateral line system. *Biological cybernetics*, 106, 627-642.

[31] Coombs, S., Bleckmann, H., Fay, R. R., & Popper, A. N. (Eds.). (2014). *The lateral line system* (pp. xiv-347). New York: Springer.

[32] Yang, Y., Klein, A., Bleckmann, H., & Liu, C. (2011). Artificial lateral line canal for hydrodynamic detection. *Applied Physics Letters*, 99(2).

Dear Editors and Reviewers:

Thank you once again for your time and insightful comments on our manuscript entitled 'An Interpretable and Generalizable Data-Driven Model for Robust Self-State Estimation of a Fishlike Robot' (NCOMMS-24-28282A). We greatly appreciate your valuable feedback, which has been instrumental in refining and enhancing our work. We have carefully considered each comment and implemented comprehensive revisions accordingly. We believe these improvements have significantly enhanced the clarity and overall quality of the manuscript. Specifically, *the comments are in black and numbered. Our answers are in blue, and the modified texts from the manuscript are in red.*

Reviewer #1 (Remarks to the Author):

R1.Q1

My comments have been properly addressed. The new experiments with non-steady state conditions (e.g. varying frequencies + turns) are good additions. Also the new videos are informative. I recommend accepting for publication.

R1.A1

Thank you for your time and effort in reviewing our manuscript. Your constructive feedback and insightful suggestions have been valuable in improving the quality and clarity of our work. Thank you again for your support and for contributing to the refinement of our research.

Reviewer #2 (Remarks to the Author):

R2.Q1

The authors have made significant effort in addressing some of my comments.

However, they still fall short to clearly address some points:

R2.A1

Thank you for your valuable and constructive feedback. In this revised version, we have carefully addressed all the comments, particularly those related to potentially misleading phrasing, such as the incorrect use of the term 'turbulence' and 'Lighthill's theory'. We sincerely appreciate your insights. We believe that the updated manuscript now provides clearer explanations of our method, as well as a balanced discussion of its strengths and limitations, which are crucial for our readers.

R2.Q2

* R2.Q4/R2/A4: "generalizable" and "other complex cases". I am sorry but I understand the meaning of the work "generalizable" but precisely, I'd like to know what kind of complex cases we're talking about.

R2.A2

Thank you for your comment. We appreciate your attention to the generalizability of the method. In the first two sections of Results, we interpret the POD modes and coefficients, use coefficients for velocity estimation and use modes for selecting the optimal combination of sensors for a swimming fishlike robot with constant oscillating parameters (10 cases). Then, in the third section of Results, we directly use this method without any modifications for 4 cases with unsteady oscillating parameters (Fig. 6) and 18 cases with interference from the vortices shedding from the leading robot (Fig. 8). Finally, we also extend our method for fishlike robots with different three-dimensional morphologies and swimming styles (Fig. 7). Our method all works well in these 'complex cases'.

Line 85: We finally demonstrate the generalizability of our method **in three complex cases**.

1) Our method works effectively for the free-swimming fishlike robot under various oscillation parameters, including varying frequencies, amplitudes, and offsets. 2) It can be extended to three-dimensional pressure data obtained through three-dimensional CFD simulations with both boxfish and eel-like models, suggesting that our method can be generalized to fishlike robots with different morphologies and swimming styles. 3) The estimation method proves robust in self-velocity estimation of the fishlike robot swimming in complex flows with vortices shedding from a neighboring robot.

R2.Q3

I can't accept that the method is truly general. There must be a number of limitations, which have to be spelt out.

R2.A3

Thank you for pointing this out. Although we have tested so many cases and demonstrated the potential generalizability of this method, we agree that we could not say it will work for other extreme cases which we haven't tested yet. A possible case is when the robot oscillates very slightly and swims very slowly (less than 0.03 m/s), where the Reynolds number is small (less than 10,000). In these cases, both inertial and viscous forces play an important role in affecting hydrodynamic characteristics. As the Reynolds number increases, inertial forces become dominant [1]. The POD method in our manuscript primarily analyzes the pressure data, which are related to inertial forces. So our method is more suitable for situations with a higher Reynolds number, generally above 10,000 (swimming velocity larger than 0.03 m/s) [1]. In addition, from the perspective of signal-to-noise ratio, when the fishlike robot moves very slowly, the pressure signals recorded by sensors are extremely weak (even weaker than the noise), making it difficult for our method to effectively decompose the pressure data due to the low data quality.

One other potential case is when the robot swims very fast (e.g. over 10 m/s, Reynolds number over 3,000,000). In such cases, the water pressure could drop below its vapor pressure, leading to cavitation and the formation of small vapor bubbles. These bubbles can disrupt the pressure measurements [2], and may even cause physical damage to the sensors. As a result, pressure sensors may fail to capture effective pressure signals and POD could not get meaningful features.

Another potential case would be when the robot swims with extreme agility, such as during a sudden C-start or S-start. These rapid movements could produce large peaks in the pressure signals, potentially causing instability and anomalies in POD results.

Our method requires further development in many other extreme scenarios. We sincerely appreciate the valuable insights you have provided, which will help guide our future research. To avoid any potential misunderstanding or overselling, we have limited the 'generalizability' of our method to the typical swimming patterns of robots/fish, which can be broadly defined by velocity ranges from 0.03 to 0.3 m/s in our manuscript. Our method should be effective as long as the pressure sensor can capture meaningful pressure data relative to noise. In such cases, there is a high likelihood that the method will perform well up to a velocity limit of at least 1 m/s.

We have added a full discussion about the limitations of our method in the revised manuscript.

Line 477: POD remains effective across the entire range of operational parameters for the fishlike robot and maybe for real fish as well. However, this method may struggle to extract meaningful pressure modes at those extreme cases, such as very low swimming velocity due to the increased influence of viscous forces⁶⁷ and the weakening of effective pressure signals, or extremely high velocity due to cavitation effects⁶⁸. All these would be our main future work to extend our method to handle these extreme and complex cases.

R2.Q4

* R2.Q5: the term "turbulence" is used in a very loose way. Again, what kind of turbulence are we talking about? Homogenous isotropic turbulence? Turbulent boundary layer? etc.

R2.A4

Thank you for the valuable feedback. We apologize for the loose use of the term 'turbulence' here, as it could easily be misleading to the reader. What we intended to convey is that our method is also capable of accurately estimating the velocity of the fishlike robot facing the interference by the vortices shedding from a preceding robot, rather than in a simple laminar flow environment. In our experiments, the leading robot generates vortices through the oscillation of its body and tail, which interfere with the pressure data collected by the sensors on the following robot. The leading fish serves merely as a source of vortex disturbance, so using 'turbulence' is not entirely accurate. We have revised this to 'flows with vortices' in our manuscript.

Line 403: To further generalize our proposed method for self-state estimation from the perspective of applications, we conduct experiments with the fishlike robot swimming in flows with vortices shedding from a preceding robot, which is a typical complex flow environment in fish schools, and use ALLS to estimate the self-velocity (detailed in the Methods section).

R2.Q5

* R2.Q7: in R2.A7, the authors state that they intend to use Lighthill's theory in 3D in the future. This is NOT trivial at all. Are they referring to the Poincaré–Cosserat equations? They would need to provide more details to be convincing.

R2.A5

Thank you for raising this point. The Lighthill's theory used in our manuscript refers to the theoretical model that predicts the hydrodynamic pressure on the surface of a swimming clupeid fish given the velocity and acceleration components of the body [3]. The pressure is expressed in Eq. (3) in our manuscript. This model can be extended to 3D as follows. If the movement of the fishlike robot is still in a 2D plane (only with U , V , ω), the 3D pressure on the surface can be expressed with 2D coordinates X , Y , together with the third coordinate Z . In addition, if the fishlike robot swims in the 3D space (with upward velocity, pitching and rolling angular velocities), we can include the 3D velocity components in Eq. (3) in our manuscript.

We are concerned that you may have been referring to Lighthill's well-known large-amplitude elongated-body theory of fish locomotion [4], which we did not use in this work. As explained above, our work utilizes Lighthill's pressure model of the swimming fish given the movement (velocity and acceleration components) [3]. We agree that extending the elongated-body theory from 2D to 3D is challenging, even though some have attempted it [5]. But extending the hydrodynamic pressure model from 2D to 3D is relatively easy as we explained above. We apologize if the term 'Lighthill's theory' caused any confusion. Thank you for highlighting this point, as it could potentially mislead future readers as well. To avoid any further misunderstanding, we have revised 'Lighthill's theory' in the manuscript to 'Lighthill's pressure model'.

Line 144: To interpret the dominant modes in POD, we establish a two-dimensional

theoretical model that describes the hydrodynamic pressure variations on the surface of a swimming fishlike robot based on Lighthill's pressure model⁵⁰ (detailed in Supplementary Note 3).

R2.Q6

* R2.Q8: this is a very important point and concern. The authors are vaguely responding to it but no mention of it is made in the manuscript. This point should be stressed as a clear limitation of the present study, and this should appear explicitly in the manuscript.

R2.A6

Thank you for highlighting this important concern. We acknowledge the need to explicitly address this point in the manuscript. In response, we have revised the manuscript to clearly state this limitation in Discussion.

We conducted a preliminary POD analysis of the velocity field around a swimming fishlike robot in a 3D CFD simulation. It can be seen in the figure below that the velocity field can also be decomposed by POD. And mode 1 represents the swimming velocity with an approximately constant value on the surface and coefficient 1 can reflect the acceleration of the robot. Mode 2 and coefficient 2 reflect the oscillation velocity. POD has also demonstrated its potential for decomposing velocity fields.

In the future, we plan to conduct a comprehensive analysis of the application of POD to both the velocity and pressure fields near the fishlike robot and real fish, aiming to extract more valuable information for underwater perception. Additionally, we will equip the fishlike robot with additional velocity and shear force sensors to further enhance the performance of ALLS, bringing it closer to the sensory capabilities of real fish.

Supplementary Fig. 24 Mode decomposition (POD) of the three-dimensional velocity field value from CFD simulations for numerical boxfish, including the flow field

visualized by isosurfaces of the Q-criterion, energy proportions, modes and coefficients. The boxfish model follows the same kinematics as the experiment with oscillation frequency = 1.8 Hz, amplitude = 30°, offset = 0°.

Line 481: Moreover, POD has also demonstrated its potential for decomposing velocity fields (detailed in Supplementary Note 12). We also plan to integrate additional shear stress or velocity sensors on the surface, as real fish do, to better accommodate natural flow conditions and enhance both data quality and diversity for POD analysis. We believe this further enhancement will bring the robot's sensory capabilities closer to those of real fish, going beyond only depending on pressure signals and estimating self-states in 2D motions⁶⁹.

R2.Q7

* Finally, there have been some attempts to use brute-force machine learning techniques (specifically artificial neural networks) to use the ALLS as a object identification tool. Although the approach taken by the authors is less data-intensive, it would be worth specifying that in the manuscript.

R2.A7

Thank you for the valuable suggestion. We acknowledge the importance of distinguishing our approach from brute-force machine learning methods. We have explicitly stated in Discussion that our method is less data-intensive and focuses on extracting interpretable modes from the pressure signals, which highlights the strengths and limitations of our method compared to others.

Line 456: Recently, neural networks have been explored for flow estimation and object identification using ALLS^{19,27,32,38}. While these methods can achieve good fitting performance in a specific problem given sufficient training data, they often require large and high-quality datasets, extensive computational resources, and may lack interpretability and generalizability. In contrast, POD leverages mode decomposition techniques to extract meaningful physical features from the pressure signals, reducing data dependency while maintaining robust performance. This not only enhances the efficiency and generalizability of our method but also provides deeper insights into the hydrodynamic sensing mechanisms that could be applicable to both robotic and biological systems.

Reviewer #3 (Remarks to the Author):

R3.Q1

The authors have adequately addressed my comments and improved the manuscript's clarity. They have also included additional results to support their statements. I have no further questions and recommend it for publication.

R3.A1

Thank you for the thoughtful review and constructive feedback throughout the revision process. Your valuable comments have significantly contributed to improving the clarity and rigor of our manuscript. We truly appreciate your recommendation for acceptance.

Reviewer #4 (Remarks to the Author):

R4.Q1

I appreciate the effort the authors have made in addressing the concerns, particularly through the additional experiments, the improved interpretability of the POD modes, and the demonstrations of sensor redundancy generalizability. These enhancements significantly strengthen the manuscript's rigor and interdisciplinary appeal.

One minor suggestion for further improvement would be to explicitly connect the POD-based ALLS approach to the biological lateral line function given the general audience nature of this journal. For instance, it would be valuable to discuss whether similar mode-like processing occurs in fish and how the observed sensor redundancy and optimal placement correspond to natural systems. Adding 1–2 sentences in the Discussion to draw this connection would further enhance the bio-inspired context of the work.

Overall, I am very pleased with the improvements, and I believe the manuscript is now nearly ready for publication.

R4.A1

Thank you for your time and effort in reviewing our manuscript. Your insightful comments and suggestions have been instrumental in refining our work, and we are grateful for your recognition of the improvements.

Thank you for your suggestion regarding the further application of the POD-based approach to biological lateral line function. Building on our existing experimental platform [6] and some preliminary results, we have added a few sentences to briefly outline how the fishlike robot model could be used to simulate real fish swimming and how our method could be applied to analyze the velocity and pressure fields around the fish. Specifically, we will explore whether these fields can be decomposed into interpretable modes. Additionally, the lateral line system of real fish may exhibit redundancy, or in other words, certain regions of the lateral line may play a dominant role in perception.

Line 471: For instance, using the existing RoboTwin platforms⁶³, we can simulate the detailed flow field around real fish by using fishlike robots that share the same morphologies and kinematics. With the pressure and velocity information, our method could identify which regions of the neuromasts may play a dominant role in hydrodynamic perception.

Reference

- [1] Gazzola, M., Argentina, M., & Mahadevan, L. (2014). Scaling macroscopic aquatic locomotion. *Nature Physics*, 10(10), 758-761.
- [2] Kubota, A., Kato, H., & Yamaguchi, H. (1992). A new modelling of cavitating flows: a numerical study of unsteady cavitation on a hydrofoil section. *Journal of fluid Mechanics*, 240, 59-96.
- [3] Lighthill, S. J. (1993). Estimates of pressure differences across the head of a swimming clupeid fish. *Philosophical Transactions of the Royal Society of London. Series B: Biological Sciences*, 341(1296), 129-140.
- [4] Lighthill, M. J. (1971). Large-amplitude elongated-body theory of fish locomotion. *Proceedings of the Royal Society of London. Series B. Biological Sciences*, 179(1055), 125-138.
- [5] Candelier, F., Boyer, F., & Leroyer, A. (2011). Three-dimensional extension of Lighthill's large-amplitude elongated-body theory of fish locomotion. *Journal of Fluid Mechanics*, 674, 196-226.
- [6] Li, L., Chao, L. M., Wang, S., Deussen, O., & Couzin, I. D. (2024). RoboTwin: A Platform to Study Hydrodynamic Interactions in Schooling Fish. *IEEE Robotics & Automation Magazine*.